EMBO
Molecular Medicine

# Anti-M1R/B6R antibody characterization and bispecific design for enhanced orthopoxvirus protection

Runchu Zhao[1,2,11], Lili Wu [2,11], Yi Zhang[1,2,11], Jianrong Ma[3,11], Dezhi Liu[1,2], Yuxuan Zheng[4], Tianxiang Kong[2,5], Renyi Ma[2], Zhengrong Gao[6,7], Yan Chai[2], Yuanlang Liu[8], Yi Tian[8], Yunxiang Xia[8], Yongzhi Hou[3], Jiahan Lu[3], Zhe Cong[3], Baoying Huang[9], Wenjie Tan [9], Jing Xue [3✉], George F Gao [2] & Qihui Wang [1,2,10✉]

## Abstract

**The global outbreak of the mpox in humans, caused by the mpox virus (MPXV), underscores the urgent need for safe and effective therapeutics. In this study, we characterized the dominant MPXV immunogens, M1R and B6R, by sequencing monoclonal antibodies (MAbs) from the immunized mice and analyzing their epitopes and functions through in vitro and in vivo assessments of binding and antiviral activities. Several broadly effective anti-M1R and anti-B6R neutralizing MAbs were identified and they exhibited enhanced antiviral effects against MPXV or vaccinia virus (VACV) when used in antibody cocktail and bispecific antibody designs. Notably, the VH-CH1 switch region-inserting format of bispecific antibodies exhibited robust protective efficacy against VACV in a mouse model. Collectively, our study characterized the epitope and functional maps of anti-M1R and anti-B6R MAbs and developed promising broad-spectrum antibody candidates for the treatment of MPXV and other orthopoxvirus infections.**

**Keywords** Mpox Virus (MPXV); M1R; B6R; Bispecific Antibody; Animal Protection
**Subject Categories** Immunology; Microbiology, Virology & Host Pathogen Interaction

## Introduction

Mpox virus (MPXV), the causative agent of mpox, is an enveloped, double-stranded DNA virus belonging to the *Orthopoxvirus* genus, Poxviridae family, which also includes vaccinia (VACV), variola (VARV) and cowpox (CPXV) viruses (Breman et al, 1980; Shchelkunova and Shchelkunov, 2022). MPXV is a zoonotic virus comprising two clades: clade I, initially circulating in central Africa, particularly in the Democratic Republic of the Congo (DRC), with an approximately mortality rate of 10%, and clade II, originally prevalent in the West Africa, with a lethality rate of less than 4% (CDC, 2025b; Koplan and Foster, 1979). In 2022, an mpox outbreak caused by a clade II variant (clade IIb) of MPXV broadly emerged in non-endemic regions, prompting the World Health Organization (WHO) declared mpox a Public Health Emergency of International Concern (PHEIC), a status maintained until May 2023. However, in 2024, a surge in mpox cases in Africa driven by a new variant (clade Ib) of MPXV, led the WHO to issue a second PHEIC declaration for mpox (WHO, 2025). As of June 30, 2025, there have been 153,961 reported mpox cases globally, resulting in 380 deaths across 137 countries (WHO, 2025).

Although three live attenuated vaccines, ACAM2000, JYNNEOS and LC16m8 have been approved for prophylaxis of mpox, these vaccines pose potential risks to vulnerable populations, particularly individuals with weakened or compromised immune systems (Deputy et al, 2023; Gong et al, 2022; Lin et al, 2024). Therefore, these vaccines have only been approved to be used for high-risk individuals but not for the general public. Alarmingly, children under the age of 15 have accounted for 70% of the reported cases in the current mpox outbreak in Africa (CDC, 2025a). Moreover, on August 15, 2024, a clinical trial conducted in the DRC revealed that tecovirimat, a promising antiviral agent, did not significantly accelerate recovery in patients infected with the clade I strain of MPXV (Lenharo, 2024; Sherwat et al, 2022). These challenges highlight the urgent need for the development of broad-spectrum and effective countermeasures to combat MPXV.

Monoclonal antibodies (MAbs) have shown significant efficacy as effective therapeutic agents in the treatment of infectious disease (Dai et al, 2023; Pan et al, 2024; Su et al, 2024; Sun et al, 2024).

[1]Institute of Physical Science and Information, Anhui University, 230039 Hefei, Anhui, China. [2]CAS Key Laboratory of Pathogen Microbiology and Immunology, Institute of Microbiology, Chinese Academy of Sciences (CAS), 100101 Beijing, China. [3]State Key Laboratory of Respiratory Health and Multimorbidity, NHC Key Laboratory of Human Disease Comparative Medicine, Institute of Laboratory Animal Science, Chinese Academy of Medical Sciences and Peking Union Medical College, 100021 Beijing, China. [4]Human Phenome Institute, Minhang Hospital, Fudan University, 201203 Shanghai, China. [5]School of Medicine, Tsinghua University, 100190 Beijing, China. [6]Shenzhen Children's Hospital, 518000 Shenzhen, China. [7]Shenzhen Institute of Advanced Technology, Chinese Academy of Sciences, 518055 Shenzhen, China. [8]Wuhan Institute of Biological Products Co., Ltd., 430207 Wuhan, China. [9]NHC Key Laboratory of Biosafety, National Key Laboratory of Intelligent Tracking and Forecasting for Infectious Diseases, National Institute for Viral Disease Control and Prevention, Chinese Center for Disease Control and Prevention, 102206 Beijing, China. [10]Medical School, University of Chinese Academy of Sciences, 101408 Beijing, China. [11]These authors contributed equally: Runchu Zhao, Lili Wu, Yi Zhang, Jianrong Ma. ✉E-mail: xuejing@cnilas.org; wangqihui@im.ac.cn

During the early stages of the COVID-19 pandemic, several neutralizing MAbs have been authorized for emergency use (Dougan et al, 2021; Shi et al, 2020; Wu et al, 2020; Yang et al, 2020). Nevertheless, the continuous mutations of SARS-CoV-2 have made the MAbs particularly vulnerable to immune escape (He et al, 2024; He et al, 2023; Huang et al, 2022). Consequently, extensive research has concentrated on the combined use of multi-epitope or multi-target antibodies, as antibody cocktails and multispecific antibodies are important strategies to prevent escape and enhance the broad-spectrum efficacy (Hansen et al, 2020; Li et al, 2022; Sun et al, 2025; Tong et al, 2024; Wu et al, 2023).

MPXV has two forms of virions, the intracellular mature virus (IMV) and the extracellular enveloped virus (EEV), each carrying several surface proteins. Extensive research based on the VACV has identified several critical immunogens, including L1R (corresponding to MPXV M1R), D8L (corresponding to MPXV E8L), H3L (corresponding to MPXV H3L) and A27L (corresponding to MPXV A29L) in the IMV, as well as B5R (corresponding to MPXV B6R) and A33R (corresponding to MPXV A35R) in the EEV (Fogg et al, 2004; Hooper et al, 2003; Hooper et al, 2004; Kong et al, 2024; Smith et al, 2002). For instance, a study investigating human antibodies that cross-neutralize poxvirus infections indicated that MAbs targeting L1R/M1R, A27L/A29L, B5R/B6R and A33R/A35R were essential for efficient protection against lethal VACV infection in a mouse model (Gilchuk et al, 2016). We recently identified two human MAbs targeting MPXV B6R that exhibited effective protection against VACV (Zhao et al, 2024). In the context of vaccine design, a study demonstrated that a vaccine containing L1R, A33R and B5R proteins provided complete protection against VACV challenge (Fogg et al, 2004). Furthermore, we designed a single-chain vaccine containing soluble A35R and M1R with or without B6R exhibited 100% protection against lethal VACV challenge (Kong et al, 2024; Wang et al, 2024). These findings underscore the critical role of antibodies induced by M1R, B6R and A35R in protective immunity. However, studies focusing on the characterization of the epitopes and functions of antibodies elicited by these antigens remain limited. Additionally, no MAbs have entered clinical trials for orthopoxvirus infections.

In this study, we aim to characterize the MAbs induced by MPXV M1R and B6R, and to develop potential antibody therapeutics for the treatment of mpox. We initially sequenced multiple MAbs targeting M1R and B6R in mice immunized with mRNA vaccines (see "Methods" for details). By evaluating the binding, neutralization and protection properties of these MAbs, we mapped the epitopes and functions and identified several excellent MAbs. Furthermore, we explored the efficacy of antibody cocktail and bispecific antibody designs, which provide potential therapeutics against mpox.

## Results

### Generation of MAbs targeting key immunogens M1R and B6R of MPXV

To generate MAbs targeting M1R and B6R of MPXV, BALB/c mice were immunized with two doses of mRNA vaccines encoding MPXV antigens, followed by a VACV challenge two weeks after the second immunization (Appendix Fig. S1A). At 60 days post-initial immunization, M1R and B6R-specific germinal center B cells were isolated from the lymph nodes of the surviving mice and subjected to sequencing to acquire B-cell receptor (BCR) sequences. As a result, 471 variable regions for heavy chain (VH) and 456 variable regions for light chain (VL) were obtained with 456 paired sequences consisting of 193 expanded clones and 263 single clones. Then, we recovered 45 antibodies covering these expanded clones, which are defined as BCRs derived from B cells sharing identical V and J genes, consistent CDR3 length and at least 85% sequence similarity within the CDR3 region for both their H and L chains (Appendix Fig. S1B). The antigen specificity of these antibodies was then characterized using enzyme-linked immunosorbent assay (ELISA). The result revealed that, except for one clone that was not expressed, 11 clones targeted M1R, 26 clones targeted B6R, but 7 clones did not bind to any of the two antigens (Appendix Fig. S1C and Appendix Table S1).

### Binding characterization of anti-M1R and anti-B6R MAbs

To investigate the binding characterization of anti-M1R and anti-B6R MAbs, we analyzed their binding epitopes and binding affinities. For the 11 anti-M1R MAbs, when using the published 7D11 (Su et al, 2007; Wolffe et al, 1995) as an indicator in the epitope competition assay, mMM1-39 and mMM1-40 exhibited complete competition, while the other nine MAbs (mMM1-4, mMM1-9, mMM1-10, mMM1-16, mMM1-19, mMM1-24, mMM1-31, mMM1-35 and mMM1-41) did not, although they generally displayed competition or partial competition with each other (Fig. 1A; Appendix Figs. S2–4). These results suggested that mMM1-39 and mMM1-40 may recognize an epitope similar to that of 7D11. In addition, mMM1-10 and mMM1-16, especially mMM1-16, showed obvious binding to M1R-peptide-16 (V112-I128) in the peptide binding assays, whereas the other MAbs exhibited no binding to any of the 32 peptides (Fig. 1B). This suggests that mMM1-10 and mMM1-16 recognize the linear epitope (V112-I128), previously identified as a linear neutralization epitope on VACV L1R (Kaever et al, 2016).

The extracellular part of B6R contains four short consensus repeat (SCR) domains and a stalk region. To map the epitope of the 21 anti-B6R MAbs, we first determined the specific domain that each MAb interacts with by expressing four B6R fragments: full-length (all extracellular part, T20-H279), SCR1-2-3-4 (all four SCRs, T20-N241), SCR1-2-3 (the N-terminal three SCRs, T20-K185) and SCR1-2 (the N-terminal two SCRs, T20-E129) (Zhao et al, 2024). ELISA results indicated that five MAbs (mMB6-1, mMB6-8, mMB6-13, mMB6-29 and mMB6-42) bound only to the full-length, suggesting they may recognize the stalk region; four MAbs (mMB6-2, mMB6-6, mMB6-32 and mMB6-44) and our previously reported hMB621 bound to the full-length and SCR1-2-3-4 but showed reduced binding to SCR1-2-3 and no binding to SCR1-2, suggesting they may recognize SCR3 and SCR4; five MAbs (mMB6-5, mMB6-15, mMB6-21, mMB6-22 and mMB6-26) bound to the full-length, SCR1-2-3-4 and SCR1-2-3 but not SCR1-2, suggesting they may recognize SCR3; seven MAbs (mMB6-3, mMB6-18, mMB6-23, mMB6-28, mMB6-30, mMB6-33 and mMB6-38) bound to all four fragments, suggesting they recognize SCR1 and SCR2 (Fig. 1C). Therefore, these 21 anti-B6R MAbs were categorized into four groups based on these results. Then, we also analyzed the epitope competition of the MAbs within each group.

**A**

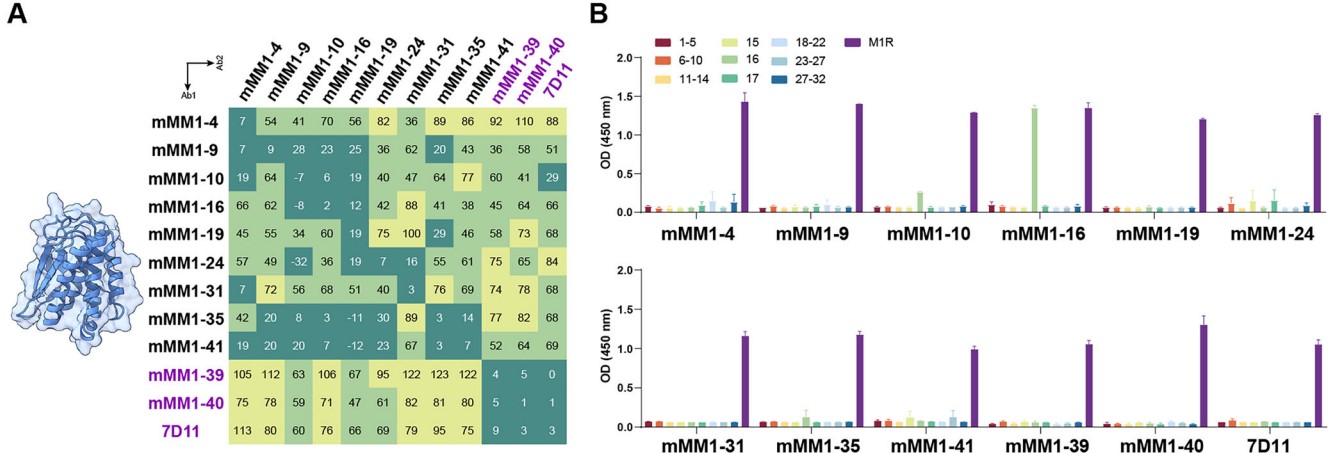

**B**

**C**

| Class | Abs | Full length (T20-H279) | SCR1-2-3-4 (T20-N241) | SCR1-2-3 (T20-K185) | SCR1-2 (T20-E129) |
|---|---|---|---|---|---|
| **Full length (T20-H279)** | mMB6-1 | √ | ✗ | ✗ | ✗ |
| | mMB6-8 | √ | ✗ | ✗ | ✗ |
| | mMB6-13 | √ | ✗ | ✗ | ✗ |
| | mMB6-29 | √ | ✗ | ✗ | ✗ |
| | mMB6-42 | √ | ✗ | ✗ | ✗ |
| **SCR1-2-3-4 (T20-N241)** | mMB6-2 | √ | √ | √ /✗ | ✗ |
| | mMB6-6 | √ | √ | √ /✗ | ✗ |
| | mMB6-32 | √ | √ | √ /✗ | ✗ |
| | mMB6-44 | √ | √ | √ /✗ | ✗ |
| | hMB621 | √ | √ | √ /✗ | ✗ |
| **SCR1-2-3 (T20-K185)** | mMB6-5 | √ | √ | √ | ✗ |
| | mMB6-15 | √ | √ | √ | ✗ |
| | mMB6-21 | √ | √ | √ | ✗ |
| | mMB6-22 | √ | √ | √ | ✗ |
| | mMB6-26 | √ | √ | √ | ✗ |
| **SCR1-2 (T20-E129)** | mMB6-3 | √ | √ | √ | √ |
| | mMB6-18 | √ | √ | √ | √ |
| | mMB6-23 | √ | √ | √ | √ |
| | mMB6-28 | √ | √ | √ | √ |
| | mMB6-30 | √ | √ | √ | √ |
| | mMB6-33 | √ | √ | √ | √ |
| | mMB6-38 | √ | √ | √ | √ |

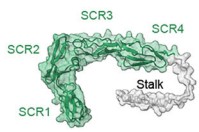
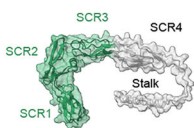
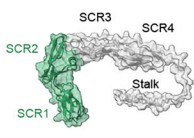

**D**

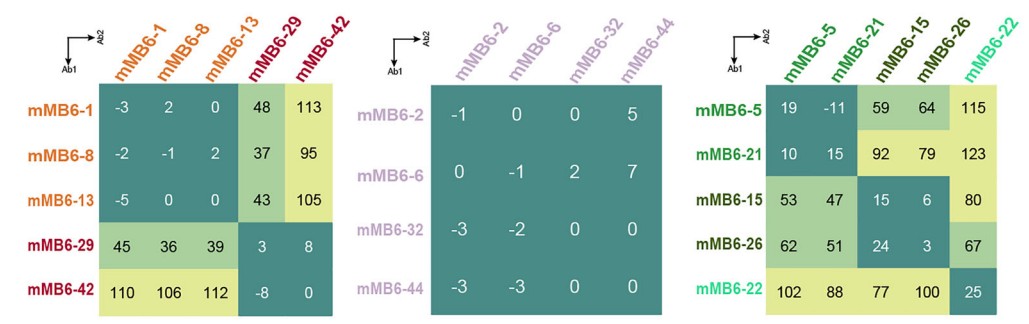

**Figure 1. Epitope classification of MAbs.**

(A) Competitive binding of 11 anti-M1R antibodies measured by bio-layer interferometry (BLI). 7D11 was used as an indicator. The structure is the MPXV M1R solved in this study, which was described in the later section. Data represent the percent of binding by Ab2 and are illustrated by colors: dark green (<30%) indicates competition, light green (31–70%) indicates partial competition, light yellow (>70%) indicates non-competition. (B) Binding of 11 anti-M1R antibodies to 32 overlapping peptides (named 1-32) spanning the ectodomain of M1R was tested by ELISA ($n = 2$). Peptides 15, 16, 17 were individually coated on the ELISA plate, while the others were coated in groups of 4–6. (C) Detection of 21 anti-B6R antibodies binding to four different B6R fragments (full length, SCR1-2-3-4, SCR1-2-3 and SCR1-2) using ELISA ($n = 2$). hMB621 was used as a control. The structure of B6R was predicted by AlphaFold. The binding of antibodies to each B6R fragment was shown. √, binding; ×, no binding; √/×, weak binding. (D) Based on the classification of anti-B6R antibodies in (C), further epitope competition within groups was assessed using BLI, as shown in (A). Source data are available online for this figure.

The full-length and SCR1-2 groups could form two subgroups, the SCR1-2-3 group split into three, while the SCR1-2-3-4 group remained a single group (Fig. 1D; Appendix Figs. S5–6).

Further binding affinity tests indicated that most anti-M1R MAbs, except for mMM1-24, exhibited high binding to M1R with affinities (equilibrium dissociation constant, $K_D$) below 50 nM (Fig. 2A; Appendix Fig. S7). Notably, mMM1-31 and mMM1-35 showed even stronger binding, with no detectable dissociation from M1R (kd < $1 \times 10^{-5}$ 1/s). Additionally, All anti-M1R MAbs can cross-bind to VACV L1R with affinities comparable to those to MPXV M1R. Based on sequence conservation analysis, M1R is identical across the four MPXV clades and exhibits over 98% similarity with its orthologs in VACV, VARV and CPXV (Appendix Fig. S8A). This finding suggests that these MAbs have broad binding potential to MPXV, including the currently prevalent clade Ib and clade IIb, as well as to VARV and CPXV.

The binding affinities results demonstrated that all anti-B6R MAbs, except for mMB6-29 and mMB6-33, exhibited high binding to B6R with $K_D$ values ranging from 0.44 to 60.88 nM (Fig. 2B; Appendix Fig. S9). Notably, MAbs targeting the SCR1-2-3-4 showed higher affinities. Furthermore, like the anti-M1R MAbs, these MAbs also showed broad binding to VACV. Sequence alignment analysis indicates that B6R is fully conserved across the four MPXV clades and shares over 92% similarity with its orthologs in VACV, VARV and CPXV (Appendix Fig. S8B). Thus, we hypothesize that these MAbs could broadly bind to MPXV, VACV, VARV and CPXV.

## Neutralizing potency of MAbs targeting M1R and B6R

Due to the cross-reactivity of the MAbs targeting M1R and B6R, we evaluated their neutralizing potency against VACV using a plaque reduction neutralization test (PRNT). Among the 11 anti-M1R MAbs, only mMM1-39 and mMM1-40 exhibited notable neutralizing abilities against VACV-IMV, with PRNT$_{50}$ values of 0.18 and 0.15 μg/mL, respectively (Fig. 2A; Appendix Fig. S10A). Additionally, mMM1-10 and mMM1-16, which bind to a linear neutralizing epitope, displayed partial inhibition at the tested antibody concentration range. The other MAbs showed no detectable neutralization activities. These results demonstrated a significant correlation between the neutralizing abilities of MAbs and their epitopes. Additionally, a positive correlation between binding affinity and neutralizing activity was observed in both the conformational (mMM1-39 and mMM1-40) and linear neutralizing epitope MAbs (mMM1-10 and mMM1-16).

For the anti-B6R MAbs, mMB6-18, mMB6-23 and mMB6-28 in the SCR1-2 group showed good neutralizing activity against VACV-EEV, with PRNT$_{50}$ values below 1 μg/mL (Fig. 2B; Appendix Fig. S10B). mMB6-1 and mMB6-8 in the full-length group, mMB6-15 and mMB6-

26 in the SCR1-2-3 group and mMB6-33, mMB6-38, mMB6-3 and mMB6-30 in the SCR1-2 displayed moderate neutralization, with PRNT$_{50}$ values over 1 μg/mL. Notably, our previously reported human neutralizing MAb hMB621, belonging to the SCR1-2-3-4 group, displayed potent neutralizing potency. Besides of the MAbs mentioned above, the remaining MAbs exhibited poor neutralizing activity, failing to achieve 50% inhibition even at the highest tested concentration of 100 μg/mL. Thus, although each group contains neutralizing MAbs, a positive correlation between neutralization activity and binding affinity was observed only in the SCR1-2 group, but not in the full-length, SCR1-2-3 or SCR1-2-3-4 group (Fig. EV1). In addition, although the EEV neutralizing assay included complement, the complement alone showed no neutralization activity (Appendix Fig. S11).

For further clinical application, the anti-L1R MAbs mMM1-40 was selected for humanization based on its binding and neutralizing activities. SPR results indicated that, compared to the parental MAb, the humanized mMM1-40 (denoted hmMM1-40) still showed strong binding to MPXV M1R and VACV L1R with nanomolar binding affinity (Appendix Fig. S12A). It also can effectively neutralize VACV-IMV, although its capability exhibited a slight decrease (Appendix Fig. S12B).

## Protection of anti-M1R and anti-B6R MAbs against VACV infection in vivo

To explore the protective effects of anti-M1R and anti-B6R MAbs against VACV in vivo, we selected anti-M1R mMM1-16, mMM1-40 and hmMM1-40, along with anti-B6R mMB6-1, mMB6-42, mMB6-2, mMB6-5, mMB6-26, mMB6-22, mMB6-23 and mMB6-3, based on their epitope specificities and neutralizing activities (Fig. 3A–G). The MAbs 7D11 and hMB621 were used as positive controls, while PBS and an unrelated MAb targeting SARS-CoV-2 served as negative controls. Consistent with our previous studies, two doses of 10 mg/kg MAbs were administered via intraperitoneal (i.p.) route 4 h before and 4 h after an intranasal (i.n.) challenge of a lethal dose of VACV (Fig. 3A). The result showed that mice in the negative control MAb and PBS groups exhibited sustained weight loss throughout the monitoring period, with weight reduction exceeding 20% at 5 days post-infection (dpi) (Fig. 3B,E). Mice treated with mMM1-16, mMM1-40 and hmMM1-40 displayed an initial weight loss followed by regain and a subsequent decline, similar to the 7D11 group. Notably, weight loss in all anti-M1R MAbs groups remained below 20% during entire monitoring period, with the mMM1-16 group showing the most significant weight loss at 4 dpi and 5 dpi (Fig. 3B,C). Analysis of viral loads in lungs indicated that the mMM1-40 and hmMM1-40 groups, but not the mMM1-16 group, exhibited a significant reduction compared to the PBS group, consistent with their in vitro neutralization results (Figs. 2A and 3D).

**A**

| Abs | Binding affinity ($K_D$: nM) | | Neutralization ($PRNT_{50}$: µg/ml) |
|---|---|---|---|
| | **MPXV M1R** | **VACV L1R** | **VACV-IMV** |
| mMM1-4 | 1.49 ± 0.25 | 1.74 ± 0.07 | — |
| mMM1-9 | 8.98 ± 0.32 | 4.91 ± 0.14 | — |
| mMM1-19 | 40.24 ± 3.61 | 108.60 ± 32.41 | — |
| mMM1-24 | 233.62 ± 48.36 | 265.41 ± 171.35 | — |
| mMM1-31 | # | 0.176 ± 0.08 | — |
| mMM1-35 | # | 0.11 ± 0.002 | — |
| mMM1-41 | 0.45 ± 0.06 | 0.19 ± 0.03 | — |
| mMM1-10 | 8.98 ± 0.72 | 13.67 ± 1.33 | >100 |
| mMM1-16 | 6.30 ± 0.19 | 0.72 ± 0.14 | >100 |
| mMM1-39 | 1.17 ± 0.26 | 1.77 ± 0.24 | 0.18 |
| mMM1-40 | 0.09 ± 0.04 | 0.07 ± 0.01 | 0.15 |

**B**

| Class | Abs | Binding affinity ($K_D$: nM) | | Neutralization ($PRNT_{50}$: µg/ml) |
|---|---|---|---|---|
| | | **MPXV B6R** | **VACV B5R** | **VACV-EEV** |
| Full length (T20-H279) | mMB6-1 | 1.78 ± 0.55 | 9.23 ± 1.21 | 4.89 |
| | mMB6-8 | 8.98 ± 3.28 | 14.80 ± 4.47 | 8.6 |
| | mMB6-13 | 8.93 ± 2.72 | 28.50 ± 5.27 | >100 |
| | mMB6-29 | 390.40 ± 32.94 | 116.00 ± 65.10 | >100 |
| | mMB6-42 | 0.51 ± 0.05 | 0.99 ± 0.11 | >100 |
| SCR1-2-3-4 (T20-N241) | mMB6-2 | 5.88 ± 2.52 | 8.15 ± 1.31 | >100 |
| | mMB6-6 | 1.03 ± 0.74 | 172.00 ± 19.40 | >100 |
| | mMB6-32 | 0.44 ± 0.08 | 3.93 ± 4.58 | >100 |
| | mMB6-44 | 1.94 ± 0.11 | 5.80 ± 0.54 | >100 |
| | hMB621 | 1.90 ± 1.20 | 2.78 ± 1.65 | 0.06 |
| SCR1-2-3 (T20-K185) | mMB6-5 | 12.67 ± 5.33 | 18.30 ± 1.07 | >100 |
| | mMB6-21 | 56.40 ± 32.87 | 17.50 ± 6.90 | >100 |
| | mMB6-15 | 1.78 ± 0.55 | 20.20 ± 4.98 | 2.07 |
| | mMB6-26 | 2.49 ± 0.93 | 7.08 ± 1.21 | 1.65 |
| | mMB6-22 | 46.23 ± 10.81 | 4.64 ± 0.69 | >100 |
| SCR1-2 (T20-E129) | mMB6-18 | 5.08 ± 1.46 | 17.20 ± 1.83 | 0.27 |
| | mMB6-23 | 4.44 ± 1.21 | 7.85 ± 0.39 | 0.25 |
| | mMB6-28 | 2.79 ± 1.37 | 4.40 ± 0.56 | 0.63 |
| | mMB6-33 | 108.86 ± 30.55 | 146.00 ± 31.30 | 12.03 |
| | mMB6-38 | 38.95 ± 13.27 | 23.40 ± 1.04 | 2.17 |
| | mMB6-3 | 25.41 ± 7.79 | 35.80 ± 4.27 | 9.54 |
| | mMB6-30 | 60.88 ± 59.00 | 21.50 ± 14.80 | 10.84 |

◄ **Figure 2. Binding and neutralizing abilities of MAbs.**

(A, B) Binding affinities ($K_D$) of 11 anti-M1R mAbs to MPXV M1R and VACV L1R (A) and 21 anti-B6R mAbs to MPXV B6R and VACV B5R (B) were measured by surface plasmon resonance (SPR). The assays were repeated three times. Shown data were the mean ± SD of three independent experiments. Neutralizing potencies of anti-M1R MAbs against VACV-IMV (A) and anti-B6R MAbs against VACV-EEV (B) were tested using a plaque reduction neutralization test (PRNT) ($n = 2$). Representative PRNT$_{50}$ values of two independent experiments are shown. Source data are available online for this figure.

For the anti-B6R MAbs, the mMB6-1, mMB6-26 and mMB6-23 groups demonstrated weight regain or maintenance after 3 dpi (Fig. 3E,F). The mMB6-26 and mMB6-23 groups exhibited trends similar to hMB621, with all mice surviving, while the mMB6-1 group exhibited a 60% survival rate. Mice in the other MAb groups (mMB6-42, mMB6-2, mMB6-5, mMB6-22 and mMB6-3) continued to lose weight throughout the monitoring period. Lung viral load analysis revealed that the mMB6-26 and mMB6-23 groups had significantly lower viral loads compared to the PBS group, similar to the hMB621 group. Notably, the reduction in lung viral load in these two groups surpassed that observed in the mMM1-40, hmMM1-40 and 7D11 groups (Fig. 3D,G). Although the mMB6-1 and mMB6-3 groups also showed reduced viral loads, their reductions were less than those seen in the mMB6-26 and mMB6-23 groups. The mMB6-42, mMB6-2, mMB6-5 and mMB6-22 groups did not exhibit a decrease in viral load, consistent with their in vitro neutralizing results (Figs. 2B and 3G).

Based on the preliminary findings, we further conducted an extended observation of the protective efficacy of the MAbs over 9 days. The results indicated that the mMM1-40 and hmMM1-40 groups exhibited weight decrease from 4 dpi to 7 dpi, followed by an increase after 7 dpi, with no mortality observed (Fig. 3H,I). However, the mMM1-16 group died at 6 dpi despite surviving until 5 dpi. Lung viral load analysis revealed that the mMM1-40 and hmMM1-40 groups had significantly lower viral loads compared to the PBS group, approaching the detection limit (Fig. 3J). For the anti-B6R MAbs, the body weights of mice treated with the mMB6-1, mMB6-26 and mMB6-23, continued to increase after 2 dpi (Fig. 3K). All mice in the mMB6-26 and mMB6-23 groups survived, while the survival rate in the mMB6-1 group was 80% (Fig. 3L). Lung viral load assessments showed that the mMB6-1, mMB6-26 and mMB6-23 groups had significantly lower viral load compared to the PBS group, with viral load approaching the minimum detection limit (Fig. 3M). Notably, although the lung viral load in mice receiving the anti-M1R MAbs was not as low as that in the anti-B6R MAb groups at 5 dpi, they were close to the minimum detection line at 9 dpi (Fig. 3D,J,G,M). These results suggest that the anti-M1R hmMM1-40, as well as the anti-B6R mMB6-26 and mMB6-23, exhibit effective antiviral potential against VACV infection in a mouse model. Our previously published human MAb hMB621 also displayed comparable in vivo antiviral abilities.

## Molecular basis of the hmMM1-40 and mMM1-16 binding to M1R

To elucidate the molecular basis underlying the potent neutralizing and protective activity of hmMM1-40, we determined the crystal structure of the hmMM1-40–MPXV M1R complex at 2.80 Å resolution (Fig. 4A,B; Appendix Table S2).

During the initial sequence alignment, we found that the heavy (H) chain variable region of hmMM1-40 shares 78.99% amino acid identity with that of the previously reported L1R-targeting neutralizing antibody 7D11 (Fig. 4G). Despite this moderate sequence similarity, both antibodies originate from the same V gene germline and contain identical numbers of residues in their HCDR3 loops.

Structural analysis revealed that the H chain of hmMM1-40 contributes the vast majority of contacts with MPXV M1R (255 out of 259), similar to 7D11's interaction with VACV L1R (236 out of 239) (Fig. 4A,F; Appendix Table S3). Specifically, HCDR1, HCDR2, and HCDR3 of hmMM1-40 contribute 77, 89, and 77 van der Waals contacts and 4, 7, and 4 hydrogen bonds, respectively. Likewise, the corresponding regions of 7D11 contribute 82, 101, and 49 van der Waals contacts and 3, 6, and 2 hydrogen bonds, respectively. The HCDR1 regions differ at only one position (residue 28), where hmMM1-40 contains isoleucine and 7D11 contains threonine; however, this residue is not involved in antigen binding. Their HCDR2 regions are identical in sequence and mediate the majority of antigen interactions in both antibodies. In contrast, HCDR3 sequences differ at 7 out of 12 positions. Of these, four residues (positions 99, 100, 105, and 107) do not participate in antigen binding. The remaining three residues (101-103) form significantly more interactions in hmMM1-40 (18 van der Waals contacts and 2 hydrogen bonds at position 101; 26 and 1 at position 102; and 7 van der Waals contacts at position 103) than in 7D11 (3 and 1; 13 and 1; and 3, respectively).

To further characterize the critical interactions of hmMM1-40 and 7D11 with M1R or L1R, we performed alanine-scanning mutagenesis on selected M1R residues that form hydrogen bonds with hmMM1-40 and overlap with the shared binding interface of 7D11, including E25, Q31, T32, K33, D35, D60, D62, K125, and K127 (Fig. EV2A,B). Alanine substitutions at K33, D35, and D60 impaired the binding of hmMM1-40, hmMM1-39, and 7D11 to varying degrees, with the most pronounced effect observed in 7D11. Notably, the D35A mutation almost completely abolished the binding of all three antibodies, consistent with previous findings regarding 7D11 (Su et al, 2007; Zeng et al, 2023).

In addition, we also resolved the crystal structure of the mMM1-16 in complex with M1R at a resolution of 2.70 Å (Fig. 4C,D; Appendix Table S4), representing the first structural characterization of the linear neutralizing epitope on M1R. Structural analysis revealed that mMM1-16 binds to the Q114-L126 region on M1R (Fig. 4E; Appendix Table S5), consistent with the peptide binding result (Fig. 1B). The CDR1, CDR2, FR3 and CDR3 in the H chain and the CDR1 and CDR3 in the L chain of mMM1-16 participate in the interaction with M1R. W33 (H-CDR1), D55 (H-CDR2), T101 (H-CDR3), H30 (L-CDR1) and W92, T94 and Y96 (L-CDR3) form H-bonds with M1R (Fig. 4D).

## Protection of MAbs and antibody cocktail against MPXV infection in vivo

Based on the effective in vivo anti-VACV abilities of hmMM1-40 and hMB621, we further evaluated their in vivo antiviral efficacy, including

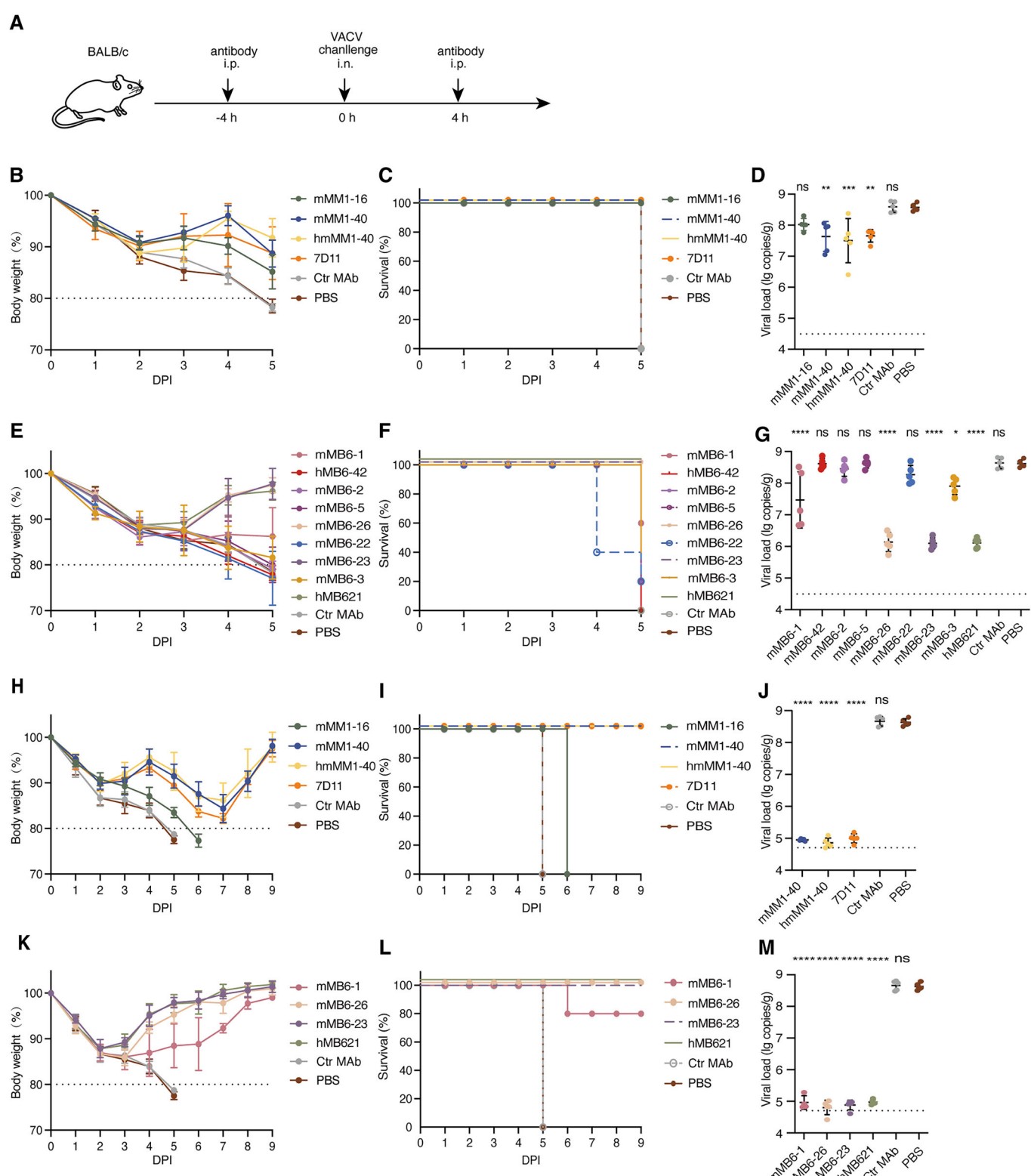

synergistic effects against MPXV. Experiment was performed on CB-17 SCID mice (Osorio et al, 2009; Wei et al, 2023). The results indicated that the hmMM1-40 and hMB621+hmMM1-40 cocktail groups exhibited continuous weight increase, with no mortality observed during the 10-day period (Fig. 5A,B). However, the hMB621

group showed a weight decrease at 5 dpi, resulting in an 80% survival rate. The PBS group experienced a decrease in body weight and developed clinical symptoms at 5 dpi, including ocular inflammation, blindness, pox and disheveled fur. Similarly, the negative control MAb group showed weight loss starting at 4 dpi and displayed clinical

**Figure 3. Protection of anti-M1R and anti-B6R MAbs against VACV infection in vivo.**

(A) Schematic representation of viral challenge and antibody administration. BALB/c mice ($n = 5$) were treated intraperitoneally (i.p.) with 10 mg/kg indicated antibody or PBS 4 h before and 4 h after intranasal (i.n.) challenge with a lethal dose of VACV WR strain. **(B–M)** Body weight (**B, E, H, K**), survival (**C, F, I, L**) and viral load in lung (**D, G, J, M**) of mice. Ctr MAb, a MAb against SARS-CoV-2. Viral titers in the lungs at 5 days post-infection (dpi) in mice treated with anti-M1R antibodies are shown in (**D**). $P = 0.1152$ (mMM1-16 vs. PBS), $P = 0.0030$ (mMM1-40 vs. PBS), $P = 0.0007$ (hmMM1-40 vs. PBS), $P = 0.0036$ (7D11 vs. PBS), $P > 0.9999$ (Ctr MAb vs. PBS). Viral titers in the lungs at 5 dpi in the mice treated with anti-B6R antibodies are shown in (**G**). $P < 0.0001$ (mMB6-1 vs. PBS), $P > 0.9999$ (mMB6-42 vs. PBS), $P = 0.9984$ (mMB6-2 vs. PBS), $P > 0.9999$ (mMB6-5 vs. PBS), $P < 0.0001$ (mMB6-26 vs. PBS), $P = 0.6585$ (mMB6-22 vs. PBS), $P < 0.0001$ (mMB6-23 vs. PBS), $P = 0.0238$ (mMB6-3 vs. PBS), $P < 0.0001$ (hMB621 vs. PBS), $P > 0.9999$ (Ctr MAb vs. PBS). Viral titers in the lungs at 9 dpi in mice treated with anti-M1R antibodies are shown in (**J**). $P < 0.0001$ (mMM1-40 vs. PBS), $P < 0.0001$ (hmMM1-40 vs. PBS), $P < 0.0001$ (7D11 vs. PBS), $P = 0.9811$ (Ctr MAb vs. PBS). Viral titers in the lungs at 9 dpi in the mice treated with anti-B6R antibodies are shown in (**M**). $P < 0.0001$ (mMB6-1 vs. PBS), $P < 0.0001$ (mMB6-26 vs. PBS), $P < 0.0001$ (mMB6-23 vs. PBS), $P < 0.0001$ (hMB621 vs. PBS), $P = 0.9976$ (Ctr MAb vs. PBS).The Ctr MAb and PBS groups in J and M were sampled at 5 dpi. Dotted lines in (**D, G, J, M**) indicate limit of detection (LOD) for the assay. Data are mean ± SD ($n = 5$ mice per group). Statistical analysis was performed by using ordinary one-way ANOVA with Dunnet's multiple comparisons test. *$P < 0.05$, **$P < 0.01$, ***$P < 0.001$, ****$P < 0.0001$, and ns $P > 0.05$. Source data are available online for this figure.

symptoms at 7 dpi. Viral load analysis revealed that, compared to the PBS group, the hMB621 group exhibited reduced viral loads in the lung, spleen and ovary, although the reductions in the lung and spleen were not significant (Fig. 5C). In contrast, the hmMM1-40 group displayed significantly reduced viral loads in the lung and spleen but not in the ovary. Notably, the hMB621+hmMM1-40 cocktail group showed significantly lower viral loads in all three tissues.

Based on these results, we further investigated whether the hMB621+hmMM1-40 cocktail could reduce viral loads rapidly over a shorter period of 4 days. As expected, the hMB621+hmMM1-40 group showed a similar increase in body weight as observed previously (Fig. EV3A,B). Viral load analysis indicated that the hMB621+hmMM1-40 group had reduced viral loads in the lung, spleen and ovary, with a significant reduction in the spleen compared to the control group (Fig. EV3C). These findings suggest that the hMB621+hmMM1-40 cocktail antibody could provide effective protection against MPXV in vivo.

To further evaluate the viral clearance efficacy of the hMB621+hmMM1-40 cocktail under relatively normal immune conditions in vivo, a MPXV challenge experiment was conducted in CAST/Eij mice (Fan et al, 2025). Contrast with the observations in CB-17 SCID mice, the control MAb group exhibited a continuous increase in body weight throughout the monitoring period (Fig. 5D). Although the hMB621+hmMM1-40 group showed slight weight loss, there is no significant difference compared to the control group. Viral load tests demonstrated that the hMB621+hmMM1-40 cocktail reduced viral loads in the spleen, lung and ovary or testicle, although the reductions in the lung and ovary or testicle were not significant. Furthermore, the hMB621+hmMM1-40 significantly reduced the viral loads in liver, kidney and brain tissues (Fig. 5E).

## Protection of bispecific antibodies against VACV infection in vivo

Building on the protective effects of the hMB621+hmMM1-40 cocktail, we further designed bispecific antibodies (bsAbs) and evaluated their antiviral efficacy against VACV in vivo. According to our previous studies, we designed two formats of bsAbs, IgG-scFv format (Li et al, 2022) and VH-CH1 switch region-inserting format (Pieper et al, 2017; Wu et al, 2023), with four bsAbs generated: hMB621-hmMM1-40scFv, hmMM1-40-hMB621scFv, hMB621-sw-hmMM1-40 and hmMM1-40-sw-hMB621 (Fig. 6A). To evaluate their protective efficacy, we employed the same experimental protocol

as described in Fig. 3A, but the antibody dose was reduced to 5 mg/kg. The results revealed that the hMB621, hmMM1-40, their cocktail and bsAbs could maintain the body weight of mice during the first 4 days, with no mortality observed in the hMB621, hMB621+hmMM1-40, hmMM1-40-hMB621scFv, hMB621-sw-hmMM1-40 and hmMM1-40-sw-hMB621 groups during the 5-day monitoring period (Fig. 6B,C). However, the hmMM1-40 and hMB621-hmMM1-40scFv groups exhibited a downward trend in body weight by 3 dpi, with a survival rate of 80% at 5 dpi. Lung viral load analysis indicated that all experimental groups exhibited significantly reduced viral loads compared to the PBS group (Fig. 6D). Notably, the two VH-CH1 switch region-inserting configurations, hMB621-sw-hmMM1-40 and hmMM1-40-sw-hMB621, achieved significantly lower viral loads than the cocktail group. Conversely, the IgG-scFv configurations, hMB621-hmMM1-40scFv and hmMM1-40-hMB621scFv, exhibited higher lung viral loads than the cocktail and even the individual monoclonal antibodies. This observation is consistent with a recent study (Qu et al, 2025). Our data suggest that bsAbs targeting M1R and B6R can provide effective protection against VACV in vivo, and their efficacy is influenced by the format of bsAbs.

To further assess the protective efficacy of antibody over an extended administration window, we performed in vivo experiments in which mice were administered antibodies either 24 h before or 24 h after viral challenge, representing prophylactic and therapeutic settings, respectively (Fig. EV4). Three antibody groups were evaluated, namely the cocktail (hmMM1-40 + hMB621) and two bsAbs (hmMM1-40-sw-hMB621 and hMB621-sw-hmMM1-40). Each antibody group was administered i.p. at a dose of 5 mg/kg, and mice were monitored for 9 days post-infection. All three antibody groups provided complete protection and significantly reduced lung viral loads. In the prophylactic group (Fig. EV4A–C), the two bsAbs groups showed slightly higher residual lung viral loads compared to the cocktail group. In the therapeutic group (Fig. EV4D–F), lung viral loads of all three groups reduced to the limit of detection.

## Discussion

Neutralizing antibodies and small-molecular drugs are key therapeutics for antiviral treatment (Shen et al, 2024; Wu et al, 2024). Although several small-molecular drugs originally developed for smallpox, such as tecovirimat and brincidofovir, are available for mpox, their clinical efficacy for mpox in humans remains limited (Group et al, 2025; Titanji et al, 2024). Vaccinia immune

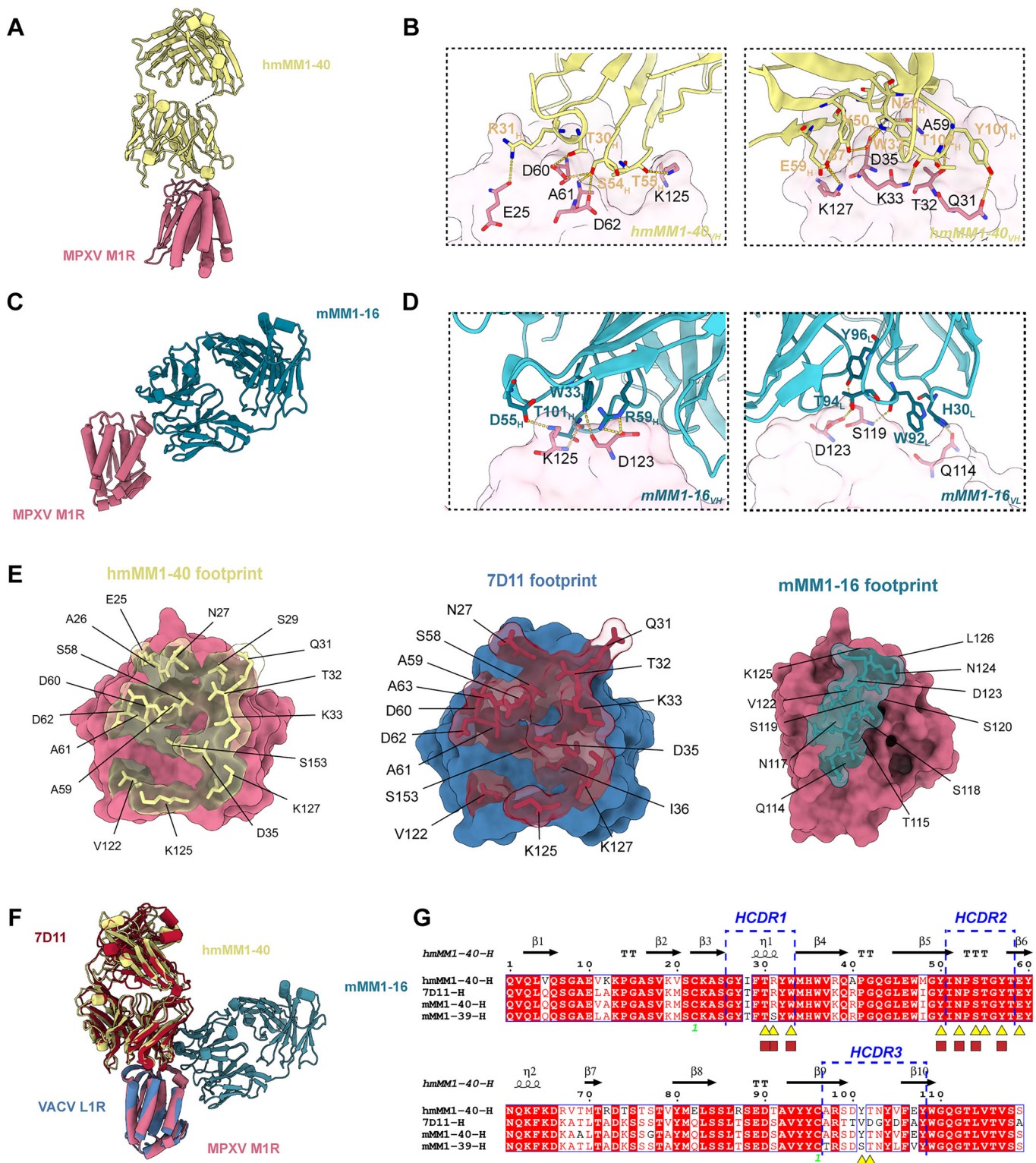

globulin (VIG), derived from individuals who received smallpox vaccine, has been used for prevention and treatment of smallpox and vaccine-related complications; however, it is in short supply. Although several neutralizing MAbs targeting critical immunogens such as M1R, B6R and A35R have been identified against MPXV

(Matho et al, 2015; Qu et al, 2025), all remain at the preclinical stage. In this study, we focused on the M1R and B6R for the development of antibody therapeutics against MPXV. We dissected the characteristics of antibodies induced by M1R and B6R from functional and structural perspectives and identified promising

**Figure 4. Structural characterization of hmMM1-40 and mMM1-16 in complex with MPXV M1R.**

(A) Overall structure of the MPXV M1R in complex with hmMM1-40 Fab. MPXV M1R is shown in pink, and hmMM1-40 Fab is shown in yellow. (B) Close-up view of the interaction between MPXV M1R and hmMM1-40 Fab. The Fab is shown in cartoon representation, and the M1R is displayed as a surface. Hydrogen bonds are indicated by yellow dashed lines, with interacting residues shown as sticks. (C) Overall structure of the MPXV M1R in complex with mMM1-16 Fab. MPXV M1R is shown in pink, and mMM1-16 Fab is shown in cyan. (D) Close-up view of the interaction between MPXV M1R and mMM1-16 Fab, with representations as in (B). (E) Epitope footprints of hmMM1-40 and mMM1-16 on MPXV M1R, and of 7D11 on VACV L1R. Epitope residues are shown as sticks with a transparent surface. (F) Structural superposition of the MPXV M1R–hmMM1-40 Fab complex, VACV L1R–7D11 Fab complex, and MPXV M1R–mMM1-16 Fab complex. hmMM1-40 Fab, 7D11 Fab, and mMM1-16 Fab are colored yellow, red, and cyan, respectively. MPXV M1R and VACV L1R are shown in pink and blue, respectively. (G) Heavy chain sequence alignment of hmMM1-40, 7D11, mMM1-40, and mMM1-39. Yellow triangles indicate hmMM1-40–MPXV M1R binding sites; red rectangles indicate 7D11–VACV L1R binding sites. Blue dashed lines highlight the positions of HCDR1, HCDR2, and HCDR3 regions.

MAbs and designed bsAbs for the treatment of mpox. Our findings offer valuable insights and guidance for future MPXV therapeutic development, targeting M1R and B6R.

For anti-M1R MAbs, we found that the conformational neutralizing epitope MAbs (mMM1-39 and mMM1-40) exhibited strong neutralizing activity, whereas the linear neutralizing epitope MAbs (mMM1-10 and mMM1-16) exhibited poor ability, consistent with previously reported 7D11 and VMC-3 (Aldaz-Carroll et al, 2005b). Interestingly, both mMM1-39 and mMM1-40 share the same H chain germline gene as 7D11, identified as IGHV1-7 in mice, which is similar to human IGHV1-46. Structural analyses of the antibody–antigen complexes further reveal that both hmMM1-40 and 7D11 engage M1R/L1R primarily through their H chains. This striking similarity in germline usage and binding mode suggests that IGHV1-7 may have been evolutionarily selected in mice as a preferred germline for conferring effective immune responses against orthopoxvirus infections.

Although the precise neutralization mechanisms of the anti-M1R MAbs remain unclear, earlier studies have suggested that the VACV L1R may function as a receptor-binding protein (Bisht et al, 2008; Foo et al, 2009; Gray et al, 2019). We therefore hypothesize that the conformational neutralizing epitope recognized by mMM1-40 likely overlaps with the receptor-binding site, enabling neutralization by obstructing the virus-receptor interaction. In addition, the linear neutralizing epitope recognized by mMM1-16 is close to the conformational neutralizing epitope and may partially interfere with receptor binding, resulting in poor neutralization. As for the non-neutralizing MAbs, their epitopes are likely distant from the conformational neutralizing epitope and may not be exposed on the virion, rendering them incapable of neutralizing virus infection. Notably, Cheng et al recently identified a novel neutralizing epitope on the membrane-proximal region of M1R, recognized by the A129 MAb, but its precise neutralizing mechanism remains to be elucidated (Qu et al, 2025).

For anti-B6R MAbs, previous studies have identified two neutralizing epitopes located in the SCR1-2 and stalk domains, with the neutralization being complement-dependent (Aldaz-Carroll et al, 2005a; Law and Smith, 2001). Recently, we identified a SCR3-4-targeting human MAb, hMB621 (Appendix Fig. S13), exhibiting significant neutralizing activity (Aldaz-Carroll et al, 2005a). In the current study, the MAbs induced by B6R were classed into four groups (full-length, SCR1-2-3-4, SCR1-2-3 and SCR1-2). Each group contains neutralizing MAbs, and MAbs targeting the SCR1-2 displayed more notable neutralization and showed a positive correlation between neutralization and binding ability, compared to those targeting other domains. These results suggest that SCR1 and SCR2 are the key regions of B6R for vaccine and therapeutic development. However, it is particularly noteworthy that the members of *Capripoxvirus* genus, including Lumpy skin disease virus, Sheeppox virus and Goatpox virus, have B6R homologs that lack the SCR1 and SCR2 domains (https://www.ncbi.nlm.nih.gov/). Interestingly, while all mouse MAbs in SCR1-2-3-4 group demonstrated poor efficacy, the human MAb hMB621 showed potent neutralization. This discrepancy may be attributed to species-specific differences between humans and mice or other underlaying factors, which need further investigation. Based on these findings, we hypothesize that the different neutralizing activities of these MAbs, whether in the same group or different groups, may attribute to the differences in the exposure of their epitopes on the virion and the accessibility of complement components. These results further suggest that the neutralizing activity of anti-B6R MAbs is highly dependent on complement and epitope, although the mechanisms remain unclear. Future structural characterization of B6R may provide valuable insights into antibody neutralization.

As previously confirmed that antibodies and vaccines targeting both IMV and EEV of orthopoxviruses can achieve optimal preventive efficacy (Davies et al, 2007; Moss, 2011). Based on the antiviral efficacy of the anti-M1R antibody hmMM1-40 and the anti-B6R antibody hMB621, we combined them into a cocktail and assessed their in vivo protection against MPXV challenge. The results demonstrated that the antibody cocktail provided better in vivo protection than either antibody alone. Interestingly, hMB621 did not exhibit robust protective efficacy against MPXV, in contrast to its performance against VACV. This discrepancy may be attributed to potential variations in the abundance of the B6R antigen on the MPXV-EEV and the B5R on the VACV-EEV, warranting further investigation in future studies. Additionally, although hmMM1-40 effectively reduced viral loads in lung and spleen, it showed limited efficacy in ovary, the mechanisms of which need further exploration in future. Furthermore, to further assess the combined efficacy of the two antibodies while simplifying the antibody purification process for potential clinical applications, we designed several bsAbs. Notably, the VH-CH1 switch region-inserting format of bsAbs, particularly hmMM1-40-sw-hMB621, demonstrated superior viral clearance in VACV challenge models compared to the antibody cocktail and the IgG-scFv format of antibodies.

In summary, we comprehensively characterized the epitope specificity and functional activity of MAbs targeting MPXV M1R and B6R. Furthermore, we explored the synergistic effects of MAbs recognizing the two targets through antibody cocktail and bispecific antibody designs. These findings highlight the potential of M1R- and B6R- based therapeutics and provide promising candidates against MPXV.

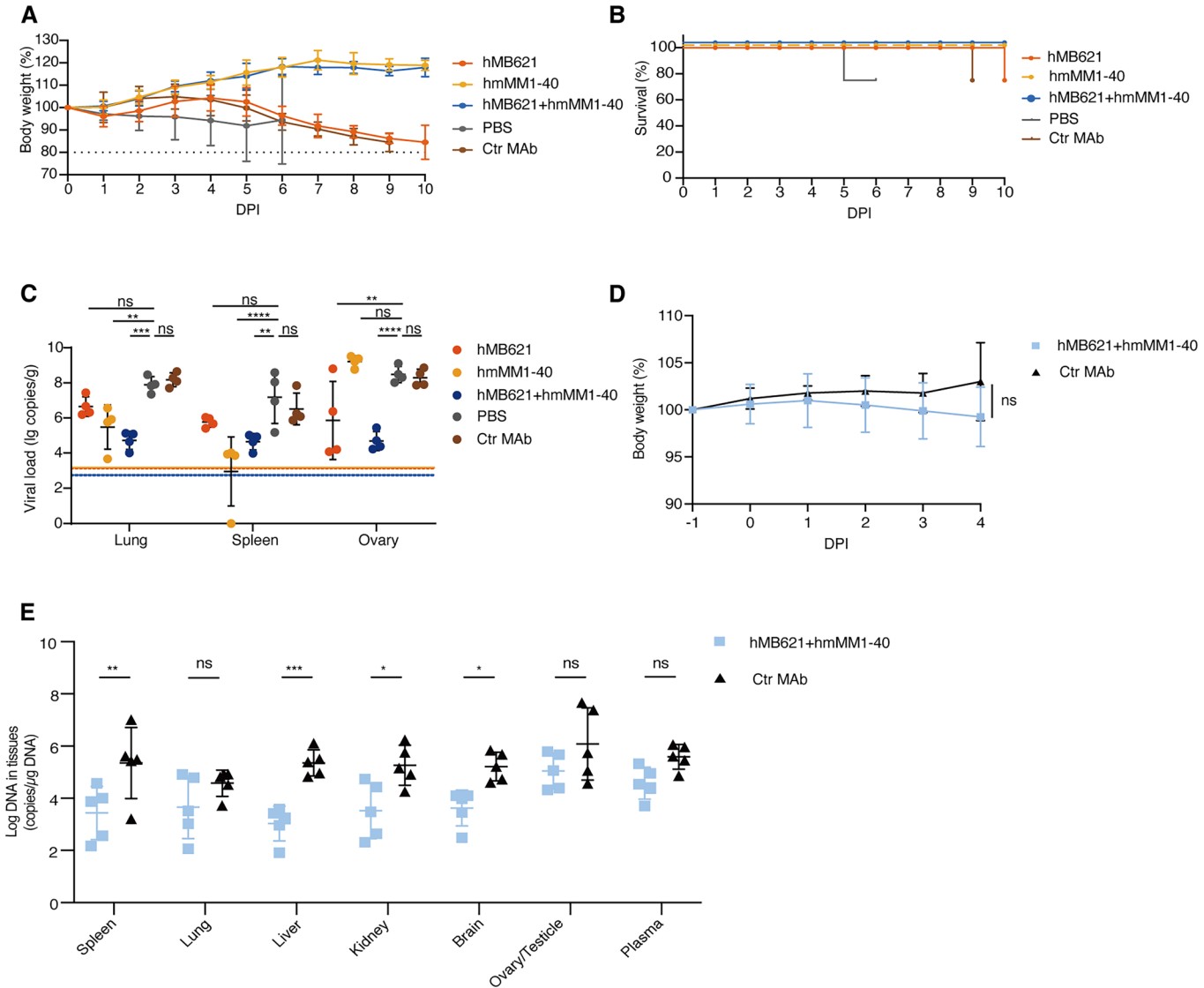

**Figure 5. Protection of antibody cocktail against MPXV infection in vivo.**

(A, B) CB-17 SCID mice were injected i.p. with 5 mg/kg of the indicated antibodies or PBS 4 h before and 4 h after i.p. and i.n. injection of a lethal dose of MPXV strain (WIBP-MPXV-001), and their body weight (A) and survival rate (B) were monitored until 10 dpi. Ctr MAb is a MAb against SARS-CoV-2. Data are mean ± SD ($n = 4$ mice per group). (C) Viral titers in the lung ($P = 0.2647$ (hMB621 vs. PBS), $P = 0.0059$ (hmMM1-40 vs. PBS), $P = 0.0003$ (hMB621+hmMM1-40 vs. PBS), $P = 0.9848$ (Ctr MAb vs. PBS)), spleen ($P = 0.1669$ (hMB621 vs. PBS), $P < 0.0001$ (hmMM1-40 vs. PBS), $P = 0.0035$ (hMB621+hmMM1-40 vs. PBS), $P = 0.7537$ (Ctr MAb vs. PBS)) and ovary ($P = 0.0027$ (hMB621 vs. PBS), $P = 0.6997$ (hmMM1-40 vs. PBS), $P < 0.0001$ (hMB621+hmMM1-40 vs. PBS), $P = 0.9996$ (Ctr MAb vs. PBS)) were quantified using quantitative real-time PCR at 10 dpi. The PBS group were sampled at 5 and 6 dpi, and Ctr MAb group was sampled at 9 dpi. The red, yellow and blue dotted lines represent the LOD for the assays of the lung, spleen and ovary, respectively. Data are mean ± SD ($n = 4$ mice per group). Statistical analysis was conducted using two-way ANOVA with Dunnett's multiple comparisons tests. *$P < 0.05$, **$P < 0.01$, ***$P < 0.001$, and ****$P < 0.0001$. (D, E) Body weight of CAST/EiJ mice were monitored following i.p. administration of 10 mg/kg indicated cocktail or Ctr MAb 24 h before i.p. challenge with MPXV strain (CAMS-CCPM-B-V-052-2306-1) and viral titers in the spleen ($P = 0.0075$ (hMB621+hmMM1-40 vs. Ctr MAb)), lung ($P = 0.5369$ (hMB621+hmMM1-40 vs. Ctr MAb)), liver ($P = 0.0007$ (hMB621+hmMM1-40 vs. Ctr MAb)), kidney ($P = 0.0192$ (hMB621+hmMM1-40 vs. Ctr MAb)), brain ($P = 0.0411$ (hMB621+hmMM1-40 vs. Ctr MAb)), ovary/testicle ($P = 0.3917$ (hMB621+hmMM1-40 vs. Ctr MAb)) and plasma ($P = 0.4208$ (hMB621+hmMM1-40 vs. Ctr MAb)) were quantified using quantitative real-time PCR at 4 dpi. Data are mean ± SD ($n = 5$, 4 male and 1 female mice per group). Statistical analysis was conducted using two-way ANOVA with Dunnett's multiple comparisons tests. *$P < 0.05$, **$P < 0.01$, ***$P < 0.001$, and ns $P > 0.05$. Source data are available online for this figure.

## Limitations of the study

Our study has some limitations. We preferentially focused on expanded clonal antibodies, whereas the single clonal antibodies may provide more epitope and function information. Additionally, although we have evaluated the antiviral efficacy of bispecific antibodies against VACV in mice, their effectiveness against MPXV should be tested in mice and even in non-human primate models in future studies.

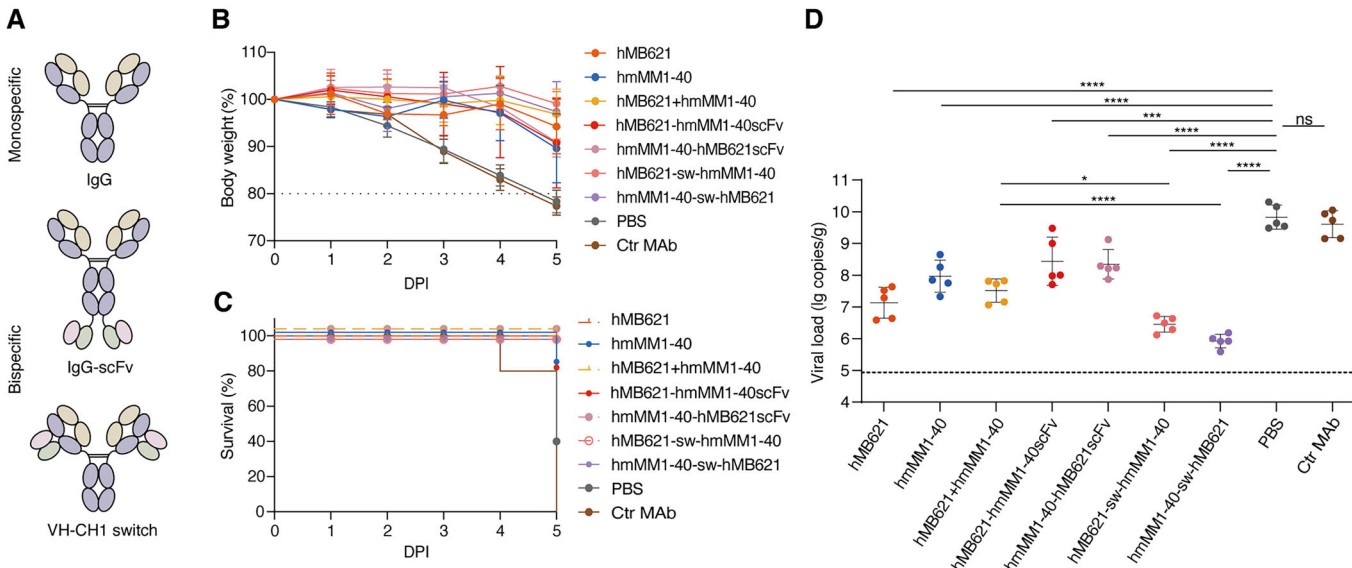

**Figure 6. Protection of bispecific antibodies (bsAbs) against VACV infection in vivo.**

(A) Schematic representation of designed bsAbs. (B, C) Body weight (B) and survival (C) of BALB/c mice were treated i.p. with 5 mg/kg indicated antibody or PBS 4 h before and 4 h after i.n. challenge with a lethal dose of VACV WR strain. Data are mean ± SD ($n = 5$ mice per group). (D) Viral titers in lungs 5 dpi were detected by using quantitative real-time PCR. Dotted line indicates LOD for the assay. $P < 0.0001$ (hMB621 vs. PBS), $P < 0.0001$ (hmMM1-40 vs. PBS), $P = 0.0002$ (hMB621-hmMM1-40scFv vs. PBS), $P < 0.0001$ (hmMM1-40-hMB621scFv vs. PBS), $P < 0.0001$ (hMB621-sw-hmMM1-40 vs. PBS), $P < 0.0001$ (hmMM1-40-sw-hMB621 vs. PBS), $P = 0.9675$ (Ctr MAb vs. PBS), $P < 0.0001$ (hmMM1-40-sw-hMB621 vs. hMB621+hmMM1-40), $P = 0.0184$ (hMB621-sw-hmMM1-40 vs. hMB621+hmMM1-40). Data are mean ± SD ($n = 5$ mice per group). Statistical analysis was performed by one-way ANOVA with multiple-comparison tests. *$P < 0.05$, ***$P < 0.001$, ****$P < 0.0001$, and ns $P > 0.05$. Source data are available online for this figure.

# Methods

### Reagents and tools table

| Reagent/resource | Reference or source | Identifier or catalog number |
|---|---|---|
| **Experimental models** | | |
| HEK293T cells | ATCC | CRL-3216 |
| Vero cells | ATCC | CL-81 |
| HeLa cells | ATCC | CCL-2 |
| HEK293F cells | Gibco | 11625-019 |
| *Escherichia coli (E. coli)* strain DH5α | TIANGEN | CB101-02 |
| *Escherichia coli (E. coli)* strain BL21 (DE3) | Novagen | 69450 |
| BALB/c mice | Beijing Vital River Laboratory Animal Technology Co., Ltd. (licensed by Charles River) | N/A |
| CB-17 SCID | Beijing Vital River Laboratory Animal Technology Co., Ltd. (licensed by Charles River) | N/A |
| CAST/EiJ mice | Institute of Laboratory Animal Science, Chinese Academy of Medical Sciences | N/A |
| VACV WR | Institute of Microbiology, Chinese Academy of Sciences | N/A |
| MPXV Clade IIb (WIBP-MPXV-001) | Wuhan Institute of Biological Products Co., Ltd. | N/A |

| Reagent/resource | Reference or source | Identifier or catalog number |
|---|---|---|
| MPXV Clade IIb (CAMS-CCPM-B-V-052-2306-1) | Institute of Laboratory Animal Science, Chinese Academy of Medical Sciences | N/A |
| **Recombinant DNA** | | |
| pCAGGS | MiaoLingPlasmid | P0165 |
| pCAGGS-MPXV B6R-His, residues T20-H279 | This paper | N/A |
| pCAGGS-MPXV M1R-His residues A3-G181 | This paper | N/A |
| pCAGGS-MPXV M1R-GFP residues A3-N250 (E25A) | This paper | N/A |
| pCAGGS-MPXV M1R-GFP residues A3-N250 (Q31A) | This paper | N/A |
| pCAGGS-MPXV M1R-GFP residues A3-N250 (T32A) | This paper | N/A |
| pCAGGS-MPXV M1R-GFP residues A3-N250 (K33A) | This paper | N/A |
| pCAGGS-MPXV M1R-GFP residues A3-N250 (D35A) | This paper | N/A |
| pCAGGS-MPXV M1R-GFP residues A3-N250 (D60A) | This paper | N/A |

| Reagent/resource | Reference or source | Identifier or catalog number |
|---|---|---|
| pCAGGS-MPXV M1R-GFP residues A3-N250 (D62A) | This paper | N/A |
| pCAGGS-MPXV M1R-GFP residues A3-N250 (K125A) | This paper | N/A |
| pCAGGS-MPXV M1R-GFP residues A3-N250 (K127A) | This paper | N/A |
| pCAGGS-VACV B5R-His residues T20-H279 | This paper | N/A |
| pCAGGS-VACV L1R-GFP residues A3-G181 | This paper | N/A |
| pET21a | Novagen | Cat# 69740 |
| pET21a-MPXV B6R-His residues T20-H279 | This paper | N/A |
| pET21a-MPXV B6R-His residues T20-N241 | This paper | N/A |
| pET21a-MPXV B6R-His residues T20-K185 | This paper | N/A |
| pET21a-MPXV B6R-His residues T20-E129 | This paper | N/A |
| pET21a-MPXV M1R-His residues A3-G181 | This paper | N/A |
| **Antibodies** | | |
| FITC Rat anti-mouse GL-7 antibody | BioLegend | 144604 |
| APC Rat anti-mouse CD138 antibody | BD Bioscience | 561705 |
| PE/Cy7 Rat anti-mouse CD38 antibody | BioLegend | 102718 |
| APC Rat anti-mouse CD93 antibody | BioLegend | 136510 |
| Brilliant Violet 421 Rat anti-mouse B220 antibody | BioLegend | 103240 |
| Brilliant Violet 510 Rat anti-mouse IgD antibody | BioLegend | 405723 |
| anti-His/PE | Miltenyi Biotec | 130-120-718 |
| Peroxidase-Conjugated Goat anti-Human IgG (H + L) | ZSGB-Bio | ZB-2304 |
| Cy3 Goat Anti-Human IgG (H + L) | APExBIo | K1208 |
| **Chemicals, enzymes and other reagents** | | |
| DMEM bASIC | ThermoFisher Scientific | C11995500BT |
| Fetal bovine serum | GIBCO | 10270106 |
| PEI | Alfa | A04043896-1g |
| SMM 293-TII | Sino Biological | M293TII |
| Sinfection-293 | Sino Biological | SF293T1 |
| QIAamp DNA Blood Mini Kit | Qiagen | 51106 |

| Reagent/resource | Reference or source | Identifier or catalog number |
|---|---|---|
| QIAamp DNA Mini Kit | Qiagen | 51306 |
| TaqMan® Gene Expression Master Mix | ThermoFisher | 4369016 |
| **Software** | | |
| FlowJo vx0.7 software | https://www.flowjo.com/ | |
| PyMOL software | https://pymol.org/2/ | |
| COOT | http://www.mrc-lmb.cam.ac.uk/personal/peemsley/coot/ | |
| Phenix | http://www.phenix-online.org/ | |
| Graphpad Prism 8.0 | https://www.graphpad.com/ | |
| Biacore Insight Evaluation, version 1.0.5.11069 | N/A | |
| UCSF Chimera | UCSF Chimera Home Page | |
| MolProbity | http://molprobity.biochem.duke.edu/index.php | |
| **Other** | | |
| HisTrap HP 5 mL column | Cytia | 17524802 |
| HiLoad 16/600 Superdex 200 pg | Cytia | 28989335 |
| Series S Sensor Chip Protein A | Cytia | 29127556 |
| Protein A HP 5 mL column | Cytia | 17040303 |
| Membrane concentrator | Millipore | UFC901096 |

## Cells and viruses

HEK293T cells (ATCC, CRL-3216), Vero cells (ATCC, CCL-81) and HeLa cells (ATCC, CCL-2) were cultured at 37 °C in Dulbecco's modified Eagle medium DMEM (Gibco, C11995500BT) supplemented with 10% fetal bovine serum (FBS) (Gibco, 10437-028). Expi293F cells (Gibco, Cat# 11625-019) were cultured at 37 °C in SMM 293-TII expression medium (Sino Biological, Cat# M293TII).

The VACV Western Reserve (VACV-WR) was obtained from Prof. Min Fang, during she worked at the Institute of Microbiology, Chinese Academy of Sciences (CAS). The MPXV clade IIb strain (WIBP-MPXV-001) was previously isolated from an mpox patient by the Wuhan Institute of Biological Products Co., Ltd. The MPXV clade IIb (CAMS-CCPM-B-V-052-2306-1) was prepared by the Institute of Laboratory Animal Science, Chinese Academy of Medical Sciences.

## Isolation of antigen-specific B cells and recombinant monoclonal antibodies production

Single-cell suspensions were prepared from the lymph nodes harvested from immunized mice and subsequently incubated with a mixture of His-tagged MPXV B6R, M1R and A35R proteins. The

cells were then stained with anti-mouse CD93-APC (1:200), CD138-APC (1:200), CD38-PE/Cy7 (1:200), IgD-BV510 (1:200), GL-7-FITC (1:200), B220-BV421 (1:200) antibodies, following the manufactures' instructions. Antigen-specific B cells gated as CD138-, CD93-, IgD-, CD38low, B220high, GL-7+ and His+ were sorted by a BD FACSAriaII flow cytometer (BD Biosciences) and subjected to high-throughput single-cell V(D)J sequencing using 10×Genomics platform. The variable regions of heavy and light chains of expanded clones were cloned into pCAGGS vector, which contains the leader sequence (METDTLLLWVLLLWVPGSTGD) and constant region of human IgG1 (for heavy chain) or Igκ (for light chain), to generate antibody expression plasmids.

## Protein expression and purification

The optimized sequences of MPXV B6R (accession no. URK20605.1), VACV B5R (accession no. YP_233069.1), MPXV M1R (accession no. URK20517.1) and VACV L1R (accession no. YP_232970.1) were fused to an N-terminal mouse Igκ signal peptide and a C-terminal 6×His tag, then cloned into the pCAGGS expression vector. The pCAGGS plasmids were transiently transfected into Expi293F cells. After 4 days, the supernatants were collected, and the soluble proteins were purified using Ni affinity chromatography with a HisTrap™ HP 5-ml column (GE Healthcare). Further purification was performed via gel filtration chromatography using a HiLoad 16/600 Superdex™ 200 Pg column (GE Healthcare) in phosphate-buffered saline (PBS).

For bacterial expression, the pET-21a plasmids containing MPXV M1R (A3-G181, accession no. URK20517.1) and MPXV B6R (T20-H279, T20-N241, T20-K185 or T20-E129, accession no. URK20605.1) were transformed into *Escherichia coli* (E. coli) strain BL21 (DE3). Following induction with 1 mM IPTG for 8 h at 16 °C, all proteins were overexpressed as inclusion bodies and subsequently refolded. After refolding, MPXV B6R and its truncated forms were exchanged in PBS buffer, while MPXV M1R was exchanged in 20 mM Tris, 150 mM NaCl (pH 8.0). The proteins were further purified by gel filtration using a HiLoad Superdex™ 75 Pg column (GE Healthcare).

Monoclonal antibodies were expressed in Expi293F cells through transient transfection. The supernatants containing the antibodies were collected and passed through a Protein A affinity column (GE Healthcare), followed by further purification using a HiLoad 16/600 Superdex™ 200 Pg column (GE Healthcare) in PBS. The mMM1-16 Fab was purified using a HisTrap™ HP 5-ml column (GE Healthcare) and a HiLoad Superdex™ 75 Pg column (GE Healthcare).

## Enzyme-linked immunosorbent assay (ELISA)

ELISA plates (Corning, 3590) were coated overnight with 2 µg/mL recombinant antigen proteins in 0.05 M carbonate-bicarbonate buffer (pH 9.6) and subsequently blocked with 5% skim milk at room temperature. Then, 10 µg/mL of purified recombinant monoclonal antibodies were added to each well at a volume of 100 µL and incubated for 1 h at room temperature. Subsequently, the plates were incubated with goat anti-human IgG-HRP antibody (ZSGB-Bio, ZB-2304) (1:4000) and developed with 3,3',5,5'-tetramethylbenzidine (TMB) substrate (Beyotime, P0209). The reactions were terminated with 2 M sulfuric acid, and the

absorbance was measured at 450 nm. The results were analyzed using GraphPad Prism 8.0.2.

## BLI assay

The Octet RED96 biosensor (Pall ForteBio) was employed to conduct competitive binding experiments with anti-M1R and anti-B6R MAbs. Briefly, streptavidin (SA) biosensors (GE Healthcare) were utilized to capture biotinylated MPXV M1R and B6R at a concentration of 10 µg/mL. Each experimental group consisted of three subgroups: subgroup 1 (M1R-first antibody-second antibody & first antibody), subgroup 2 (M1R- first antibody-first antibody), and subgroup 3 (M1R-PBST-second antibody). In subgroup 1, the sensor was exposed to the first antibody (50 µg/mL) for 500 s, followed by incubation with the second antibody (50 µg/mL) in the presence of the first antibody at the same concentration for an additional 500 s.

## SPR analysis

Initially, antibodies at a concentration of 0.5 µg/mL were captured on flow cell 2 of the Protein A sensor chip (GE Healthcare) to achieve ~500 response units (RU), while flow cell 1 served as the negative control. Following this step, serially diluted MPXV M1R, VACV L1R, MPXV B6R and VACV B5R proteins were flowed over the chip in PBST buffer. The RU measurements were conducted using a BIAcore 8 K (Cytiva) at 25 °C in single-cycle mode. After the measurements, the sensor chip was regenerated with 10 mM glycine-HCl (pH 1.5). The equilibrium dissociation constants ($K_D$) for each pair of interactions were calculated by fitting the data to a 1:1 Langmuir binding model using BIAcore® 8 K Evaluation Software (Cytiva).

## VACV IMV and EEV preparations

For VACV-IMV, a multiplicity of infection (MOI) of 0.01 of VACV-WR was inoculated into a monolayer of Vero cells at 90% confluence. After 72 h of incubation, cells were collected and rapidly frozen three times in liquid nitrogen to lyse the cells. Following centrifugation to remove cell debris, the supernatant was collected, titered, and stored at -80 °C.

For VACV-EEV, a MOI of 0.1 of VACV-WR was inoculated into a monolayer of HeLa cells. After 48 h, the medium containing the EEV form was harvested and stored at 4 °C for use within 2 weeks, as previously described (Kong et al, 2024).

## Plaque reduction neutralization test (PRNT)

PRNTs were carried out to assess the neutralization activity of anti-M1R and anti-B6R MAbs against VACV IMV and EEV, respectively. For IMV neutralization, 0.2 mL of DMEM containing approximately 150 PFU of IMV was incubated with 0.2 mL of serial two-fold dilutions of anti-M1R MAb for 1 h at 37 °C. Subsequently, Vero cell in 12-well plates were infected with the antibody-IMV mixture for 2 h at 37 °C. After infection, cells were washed twice with PBS, and a semisolid overlay consisting of 2% methylcellulose in Earle's basal minimal essential medium was then added to each well. Plates were incubated for 40 h at 37 °C. Following incubation, cells were fixed with 4% paraformaldehyde (Solarbio, P1110) at

room temperature for 1 h and subsequently stained with a 1% crystal violet solution for 1 h. Plates were then rinsed with water, and plaques were counted. The half-maximal inhibitory concentration ($IC_{50}$) values were calculated using GraphPad Prism 8.0.2.

For EEV neutralization, previously described methods were followed [31]. The assay was conducted in the presence of 10% baby rabbit complement (Cedarlane, CL3441-R) and an IMV-neutralizing anti-L1 antibody (7D11) at a concentration of 50 µg/mL. 0.2 mL of DMEM containing ~100 PFU of EEV, along with 7D11 and complement, was incubated with 0.2 mL of serial three-fold dilutions of the antibodies for 2 h at 37 °C. Subsequent experimental procedures and results analysis were consistent with those of the IMV neutralization assay.

## Crystallization and structure determination

Crystallization trials were conducted using the sitting-drop vapor diffusion method at 18 °C, by mixing 0.8 µL of protein solution with 0.8 µL of reservoir solution. Diffractable crystals of the hmMM1-40 Fab/MPXV M1R complex were obtained at the same protein concentration (8 mg/mL) in a solution composed of 1% (w/v) tryptone, 0.05 M HEPES sodium (pH 7.0), and 20% (w/v) polyethylene glycol 3,350. Similarly, crystals of the mMM1-16 Fab/MPXV M1R complex were obtained at a protein concentration of 8 mg/mL in a solution containing 0.2 M ammonium fluoride and 20% (w/v) polyethylene glycol 3350.

Diffraction data for the hmMM1-40 Fab/MPXV M1R complex were collected at beamline BL02U1 of the Shanghai Synchrotron Radiation Facility (SSRF) using a wavelength of 0.979183 Å. Diffraction data for the mMM1-16 Fab/MPXV M1R complex were collected at SSRF beamline BL17U under the same wavelength. Both datasets were processed using HKL2000 software (Otwinowski and Minor, 1997). The crystal structures were determined by molecular replacement in Phaser, using two previously reported structures (PDB: 1YPY and 2I9L) as search models. Model building was performed using Coot (Emsley and Cowtan, 2004) and structural refinement was carried out with PHENIX (version 1.20.1) (Adams et al, 2010). The stereochemical qualities of the final model was assessed with MolProbity (Williams et al, 2018). Data collection, processing, and refinement statistics are summarized in Appendix Tables S2 and 4. All structural figures were prepared using Chimera software.

## Animal protection experiments

### VACV protection in BALB/c mice
The experiments were performed in an Animal Biosafety Level 2 (ABSL-2) facility under the condition of 12 h light and dark cycle, temperature of 20–25 °C and humidity of 40-70% in Institute of Microbiology, CAS or Chinese Center for Disease Control and Prevention. All animal experiments were conducted in compliance with the guidelines and regulations of animal welfare and were approved by the Animal Ethics Committee (approval No. APIMCAS2022124).

Six- to eight-week-old female BALB/c mice were purchased from Vital River (Beijing, China) and randomly allocated to groups. Mice were given antibodies intraperitoneally (i.p.) 4 h before and 4 h after a lethal dose of VACV intranasal (i.n.) challenge. Animals were monitored daily for weight loss and survival for 5 days or

9 days following challenge. Mice that exhibited weight loss of over 20% of their initial weight were euthanized in accordance with ethical guidelines.

Magnetic Swab Viral DNA/RNA Kit (EnerTher) was used for DNA isolation of lungs. The detection of viral loads was determined with NovoStart® Probe qPCR SuperMix (UDG) Kit (Novoprotein) and Premix Ex Taq™ (Probe qPCR) (Takara) on the ABI QuantStudio 3 Real-Time PCR system.

### MPXV protection in CB-17 SCID mice
Six- to eight-week-old female CB-17 SCID mice were purchased from Vital River (Beijing, China) and randomly allocated to groups. The experiments were performed in the Animal Biosafety Level 3 (ABSL-3) facility under the condition of 12 h light and dark cycle, temperature of 20–25 °C and humidity of 40-70% in Wuhan Institute of Biological Products Co., Ltd. The study was performed in compliance with the guidelines and regulations of animal welfare, with protocols approved by the Animal Ethics Committee (approval No. WIBP-AII442023008).

Briefly, CB-17 SCID mice received antibody via i.p. injection 4 h before and 4 h after a lethal challenge dose of WIBP-MPXV-001 ($0.2 \times 10^7$ PFU intranasally and $4.8 \times 10^7$ PFU intraperitoneally). Animals were monitored daily for weight loss and survival over a period of either 4 or 10 days. Mice that lost over 20% of their initial body weight or performed severe clinical symptoms were euthanized in compliance with ethical guidelines. Viral loads in the lung, spleen and ovary were quantified using the Daan Gene Monkeypox Virus Nucleic Acid Detection Reagent (fluorescent PCR) on an ABI QuantStudio5 real-time PCR instrument.

### MPXV protection in CAST/Eij mice
MPXV protection studies in CAST/Eij mice were performed under ABSL-3 conditions under the condition of 12 h light and dark cycle, temperature of 20–25 °C and humidity of 40-70% at the Institute of Laboratory Animal Science, Chinese Academy of Medical Sciences. The experiments were conducted in compliance with guidelines approved by the Institutional Animal Care and Use Committee (approval No. XJ23002).

Four- to ten-week-old CAST/EiJ mice were divided into two groups ($n = 5$ per group), with the four in each group being male and the fifth being female. Antibodies were administered via i.p. 24 h before intraperitoneal inoculation with $1 \times 10^6$ PFU of MPXV clade IIb (CAMS-CCPM-B-V-052-2306-1) per mouse. The mice were monitored daily for weight changes. Four days post-challenge, the mice were euthanized, and plasma, spleen, lung, liver, kidney, brain, and ovary/testicle were collected. To measure viral loads, MPXV genomic DNA was extracted using a DNeasy Blood & Tissue Kit (QIAGEN). Real-time quantitative PCR (qPCR) was performed on an ABI 7500 Real-time PCR system using the TaqMan Gene Expression Master Mix (ThermoFisher).

## Statistics

Data are presented as mean ± standard deviation (SD) of the mean or as median values, as appropriate. Statistical analyses were performed using GraphPad Prism 8.0 (https://www.graphpad.com/). Two-way analysis of variance (ANOVA) was used for experiments involving multiple groups and independent variables, while one-way ANOVA was applied for analyses involving a single independent variable across

**The paper explained**

**Problem**

Mpox, caused by the mpox virus (MPXV), poses a significant threat to global public health. However, effective and specific therapeutic strategies remain lacking. Urgent efforts are needed to develop antibody-based therapeutics.

**Results**

We dissected the epitope characteristics of MPXV M1R and B6R by characterizing a panel of monoclonal antibodies (MAbs). Several broadly neutralizing anti-M1R and anti-B6R MAbs were identified and they exhibited enhanced antiviral effects against MPXV when used in antibody cocktail. Additionally, we explored bispecific antibody designs and found the VH-CH1 switch region-inserting format exhibited robust protective efficacy against VACV in a mouse model.

**Impact**

Our findings underscore the therapeutic potential of M1R- and B6R-based approaches and provide promising antibody candidates against MPXV and other orthopoxviruses.

multiple groups. Animal studies were randomized, and data collection was performed in a blinded manner. No statistical methods were used to predetermine sample size, but group sizes are comparable to those used in similar studies. All in vitro experiments were independently repeated at least twice. The exact *n* values and statistical tests used are provided in the figure legends. *P* values < 0.05 were considered significant.

# Data availability

The datasets produced in this study are available in the following database: Protein structure data: Protein Data Bank, atomic coordinates of the hmMM1-40/MPXV M1R complex and mMM1-16/MPXV M1R complex are available under accession codes 9VHZ and 9LF8, respectively.

The source data of this paper are collected in the following database record: biostudies:S-SCDT-10_1038-S44321-025-00299-z.

# Peer review information

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

## Acknowledgements

We are grateful to T Zhao Institute of Microbiology, Chinese Academy of Sciences (CAS)) for supporting the flow cytometry assay. We are grateful to Y Chen (Institute of Biophysics, CAS) and Z Fan (Institute of Microbiology, CAS)

for their technical support for SPR analysis. This work was supported by the National Key R&D Program of China (2022YFC2303400 to QW), the National Natural Science Foundation of China (82225021 to QW, 82241068 and 82222041 to JX), and the Chinese Academy of Sciences (YSBR-010 and Y2022037 to QW).

## Author contributions

**Runchu Zhao**: Data curation; Formal analysis; Validation; Writing—original draft; Writing—review and editing. **Lili Wu**: Conceptualization; Data curation; Formal analysis; Supervision; Validation; Investigation; Methodology; Writing—original draft; Writing—review and editing. **Yi Zhang**: Data curation; Formal analysis; Validation. **Jianrong Ma**: Data curation; Formal analysis; Validation. **Dezhi Liu**: Data curation; Formal analysis; Validation. **Yuxuan Zheng**: Data curation; Formal analysis; Validation. **Tianxiang Kong**: Data curation; Formal analysis. **Renyi Ma**: Data curation; Formal analysis. **Zhengrong Gao**: Data curation; Formal analysis. **Yan Chai**: Data curation; Formal analysis. **Yuanlang Liu**: Data curation; Formal analysis. **Yi Tian**: Data curation; Formal analysis. **Yunxiang Xia**: Data curation; Formal analysis. **Yongzhi Hou**: Data curation; Formal analysis. **Jiahan Lu**: Data curation; Formal analysis. **Zhe Cong**: Data curation; Formal analysis. **Baoying Huang**: Resources. **Wenjie Tan**: Resources. **Jing Xue**: Conceptualization; Data curation; Formal analysis; Supervision; Funding acquisition; Validation; Investigation; Methodology. **George F Gao**: Conceptualization; Data curation; Formal analysis; Supervision; Validation; Investigation; Methodology; Writing—original draft; Writing—review and editing. **Qihui Wang**: Conceptualization; Data curation; Formal analysis; Supervision; Funding acquisition; Validation; Investigation; Methodology; Writing—original draft; Writing—review and editing.

Source data underlying figure panels in this paper may have individual authorship assigned. Where available, figure panel/source data authorship is listed in the following database record: biostudies:S-SCDT-10_1038-S44321-025-00299-z.

## Disclosure and competing interests statement

Patents (application numbers 2023108890941 and 2024113299763) were filed containing the antibodies described in this study. QW, GFG, LW, RZ, DL and Y Zhang are the inventors. The remaining authors declare no competing interests.

# Expanded View Figures

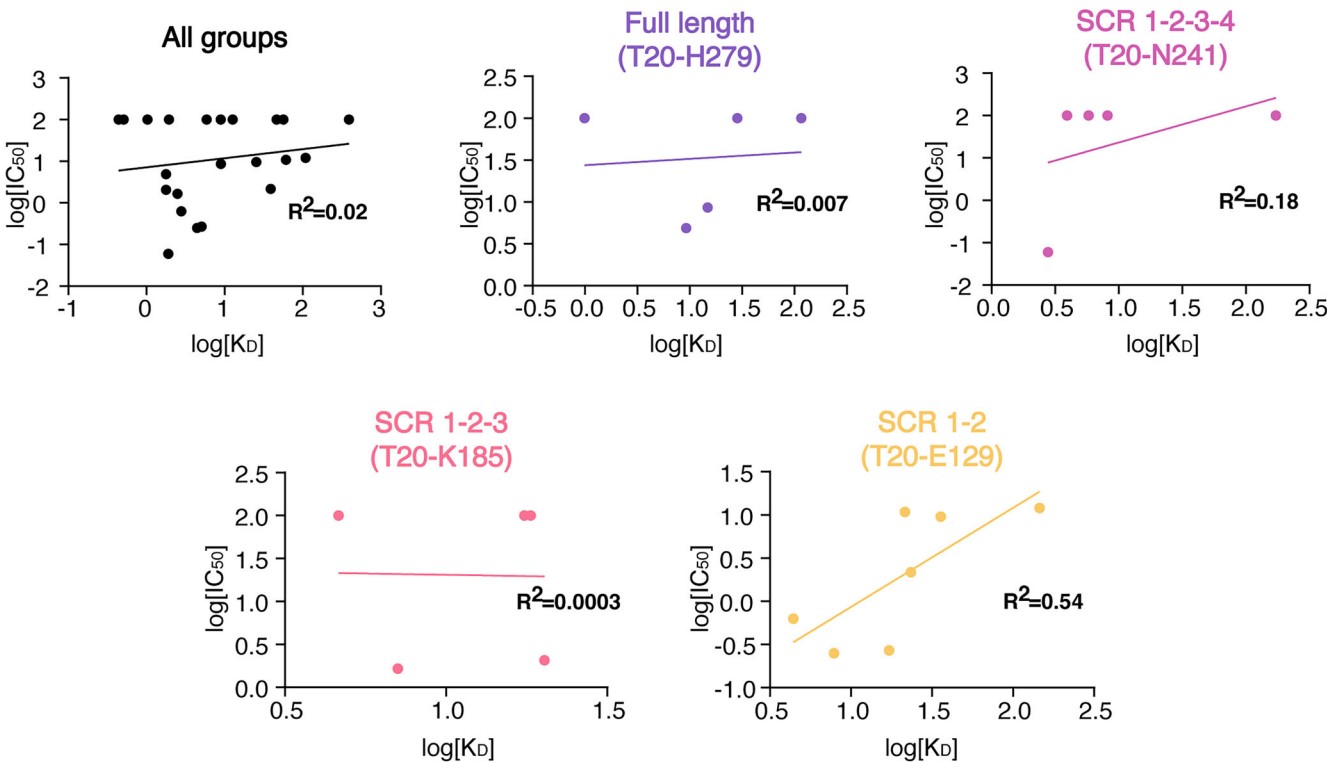

**Figure EV1. Correlation between binding and neutralization of MPXV B6R-specific MAbs.**

Log-transformed $K_D$ (VACV B5R binding) and $PRNT_{50}$ (VACV neutralization) values of epitope-specific antibodies are plotted on the horizontal and vertical axes, respectively. The straight line represents a linear regression fit, with the corresponding $R^2$ value shown nearby.

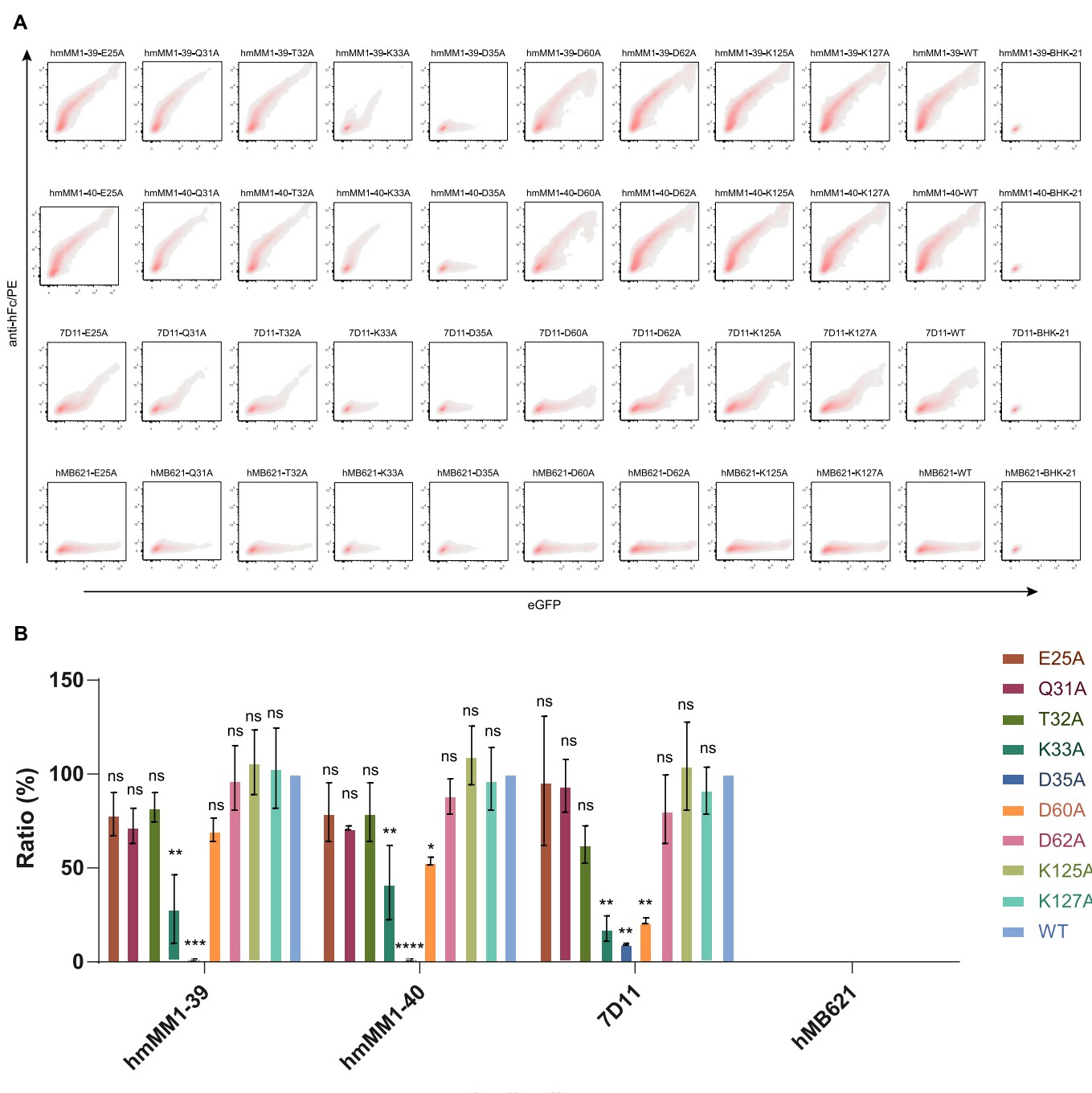

**Figure EV2. Mutational analysis of key residues in MPXV M1R involved in interactions with both hmMM1-40 and 7D11.**

(A) Density plot showing antibody binding to BHK-21 cells expressing wild-type or mutant M1R proteins. The x-axis (eGFP) indicates M1R protein expression, while the y-axis (PE) represents the level of antibody binding on the cell surface. (B) Mean fluorescence intensity (MFI) of PE within the eGFP-positive gated population, corresponding to antibody binding in cells expressing each M1R variant. In hmMM1-39 group, $P = 0.5066$ (E25A vs. WT), $P = 0.2574$ (Q31A vs. WT), $P = 0.6936$ (T32A vs. WT), $P = 0.0016$ (K33A vs. WT), $P = 0.0001$ (D35A vs. WT), $P = 0.2072$ (D60A vs. WT), $P = 0.9997$ (D62A vs. WT), $P = 0.9977$ (K125A vs. WT), $P = 0.9997$ (K127A vs. WT). In hmMM1-40 group, $P = 0.4651$ (E25A vs. WT), $P = 0.1839$ (Q31A vs. WT), $P = 0.4670$ (T32A vs. WT), $P = 0.0047$ (K33A vs. WT), $P < 0.0001$ (D35A vs. WT), $P = 0.0192$ (D60A vs. WT), $P = 0.9086$ (D62A vs. WT), $P = 0.9650$ (K125A vs. WT), $P = 0.9997$ (K127A vs. WT). In 7D11 group, $P = 0.9997$ (E25A vs. WT), $P = 0.9994$ (Q31A vs. WT), $P = 0.2051$ (T32A vs. WT), $P = 0.0028$ (K33A vs. WT), $P = 0.0014$ (D35A vs. WT), $P = 0.0039$ (D60A vs. WT), $P = 0.7775$ (D62A vs. WT), $P = 0.9997$ (K125A vs. WT), $P = 0.9941$ (K127A vs. WT). Each condition was tested in duplicate wells and the experiment was independently repeated twice. Data are mean ± SD ($n = 2$). Statistical analysis was performed by using ordinary one-way ANOVA with Dunnet's multiple comparisons test. $*P < 0.05$, $**P < 0.01$, $***P < 0.001$, $****P < 0.0001$, and ns $P > 0.05$. Source data are available online for this figure.

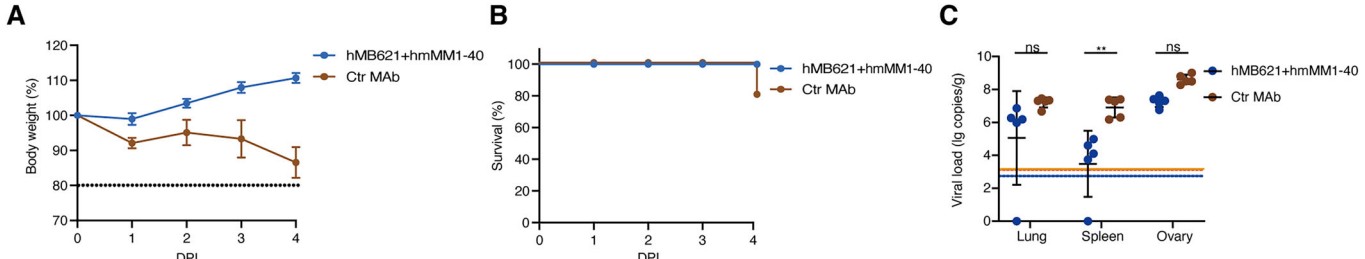

**Figure EV3.   In vivo protective efficacy of the hMB621 and hmMM1-40 antibody cocktail against MPXV infection.**

(A, B) CB-17 SCID mice were i.p. injected with a 5 mg/kg cocktail of hMB621 and hmMM1-40, or an anti-SARS-CoV-2 RBD antibody as a control (Ctr MAb), at 4 h before and 4 h after challenge with a lethal dose of MPXV strain WIBP-MPXV-001. Body weight (A) and survival (B) were monitored daily until 4 dpi. Data are mean ± SD ($n = 4$ mice per group). (C) Viral titers in the lung ($P = 0.0834$ (hMB621+hmMM1-40 vs. Ctr MAb)), spleen ($P = 0.0033$ (hMB621+hmMM1-40 vs. Ctr MAb)) and ovary ($P = 0.3980$ (hMB621+hmMM1-40 vs. Ctr MAb)) were quantified using quantitative real-time PCR at 4 dpi. The red, yellow and blue dotted lines represent the LOD for the assays of the lung, spleen and ovary, respectively. Data are mean ± SD ($n = 4$ mice per group). Statistical analysis was conducted using two-way ANOVA with Dunnett's multiple comparisons tests. **$P < 0.01$, and ns $P > 0.05$. Source data are available online for this figure.

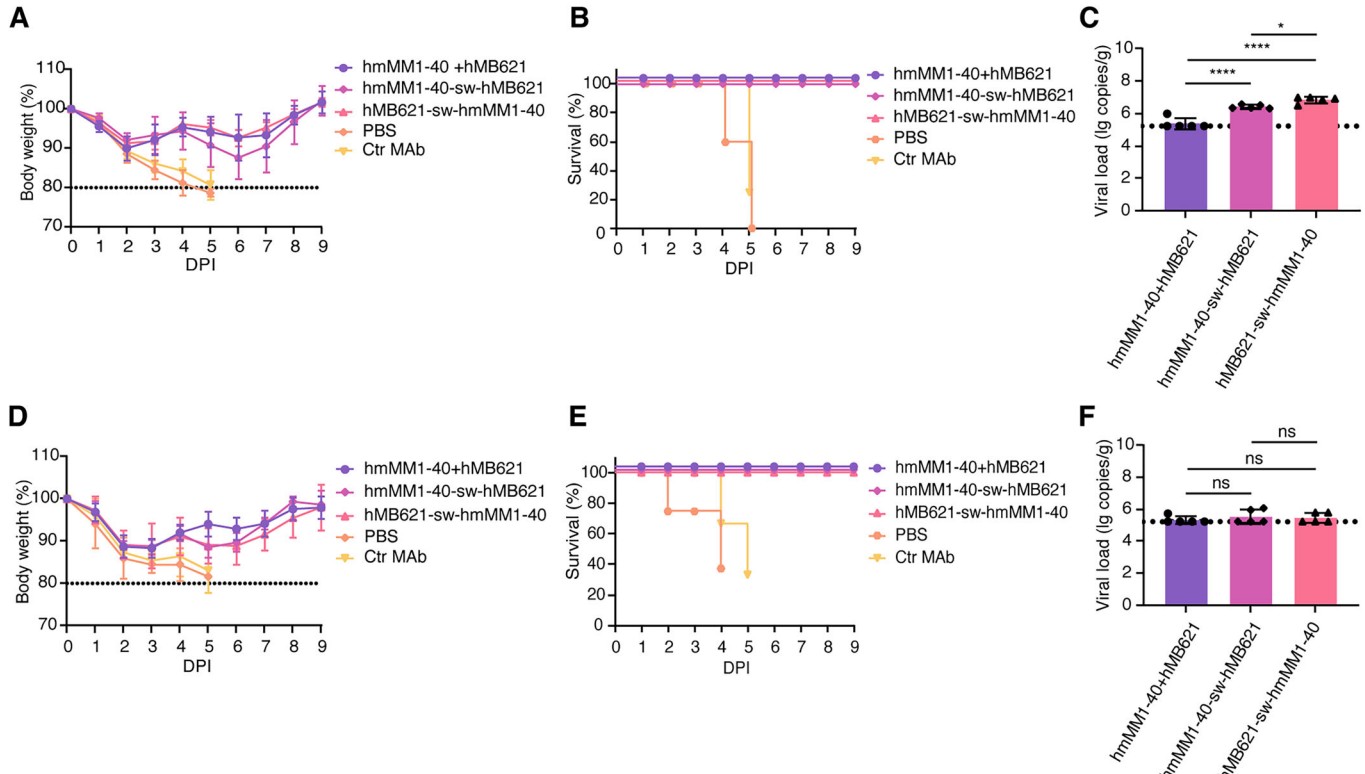

**Figure EV4. Protective efficacy of antibody treatment at −24 h and +24 h against VACV.**

(A, B, D, E). BALB/c mice ($n = 5$) received i.p. injections of 5 mg/kg of the indicated antibody or PBS either 24 h prior to (−24 h) or 24 h after (+24 h) i.n. challenge with a lethal dose of the VACV WR strain. Body weight (A, D) and survival (B, E) were subsequently monitored for 9 days. (C, F) Viral titers in the lungs at 9 dpi in mice that received antibodies 24 h before challenge, detected by quantitative real-time PCR, are shown in (C). $P < 0.0001$ (hmMM1-40 + hMB621 vs. hmMM1-40-sw-hMB621), $P < 0.0001$ (hmMM1-40 + hMB621 vs. hMB621-sw-hmMM1-40), $P = 0.0341$ (hmMM1-40-sw-hMB621 vs. hMB621-sw-hmMM1-40). Viral titers in the lungs at 9 dpi in mice that received antibodies 24 h after challenge, detected by quantitative real-time PCR, are shown in (F). $P = 0.5910$ (hmMM1-40 + hMB621 vs. hmMM1-40-sw-hMB621), $P = 0.8045$ (hmMM1-40 + hMB621 vs. hMB621-sw-hmMM1-40), $P = 0.9294$ (hmMM1-40-sw-hMB621 vs. hMB621-sw-hmMM1-40). Data are mean ± SD ($n = 5$ mice per group). Dotted line indicates LOD for the assay. Statistical analysis was performed by using ordinary one-way ANOVA with Dunnett's multiple comparisons test. *$P < 0.05$, ****$P < 0.0001$, and ns $P > 0.05$. Source data are available online for this figure.

