## [Peer Review File · EMBO Molecular Medicine]

Anti-M1R/B6R antibody characterization and bispecific design for enhanced orthopoxvirus protection

George Gao, Runchu Zhao, Lili Wu, Yi Zhang, Jianrong Ma, Dezhi Liu, Yuxuan Zheng, Tianxiang Kong, Renyi Ma, Zhengrong Gao, Yan Chai, Yuanlang Liu, Yi Tian, Yunxiang Xia, Yongzhi Hou, Jiahan Lu, Zhe Cong, Baoying Huang, Wenjie Tan, Jing Xue, and Qihui Wang

Corresponding authors: George Gao (gaof@im.ac.cn) , Qihui Wang (wangqihui@im.ac.cn), Jing Xue (xuejing@cnilas.org)

Review Timeline:

Submission Date:	6th May 25
Editorial Decision:	26th May 25
Revision Received:	6th Jul 25
Editorial Decision:	23rd Jul 25
Revision Received:	5th Aug 25
Accepted:	15th Aug 25

Editor: Zeljko Durdevic

Transaction Report:

26th May 2025

Dear Prof. Gao,

Thank you for the submission of your manuscript to EMBO Molecular Medicine. We have now received feedback from the two reviewers who agreed to evaluate your manuscript. Both referees recognize interest of the study but also raise important concerns that should be addressed in a major revision. If you would like to discuss further the points raised by the referees, I am available to do so via email or video. Let me know if you are interested in this option.

We would welcome the submission of a revised version within three months for further consideration. Please let us know if you require longer to complete the revision.

I look forward to receiving your revised manuscript.

Yours sincerely,

Zeljko Durdevic

Zeljko Durdevic
Senior Editor
EMBO Molecular Medicine

We require:

- 1) A .docx formatted version of the manuscript text (including legends for main figures, EV figures and tables). Please make sure that the changes are highlighted to be clearly visible.
- 2) Individual production quality figure files as .eps, .tif, .jpg (one file per figure). For guidance, download the 'Figure Guide PDF': (<https://www.embopress.org/page/journal/17574684/authorguide#figureformat>).
- 3) A .docx formatted letter INCLUDING the reviewers' reports and your detailed point-by-point responses to their comments. As part of the EMBO Press transparent editorial process, the point-by-point response is part of the Review Process File (RPF), which will be published alongside your paper.
- 4) A complete author checklist, which you can download from our author guidelines (<https://www.embopress.org/page/journal/17574684/authorguide#submissionofrevisions>). Please insert information in the checklist that is also reflected in the manuscript. The completed author checklist will also be part of the RPF.
- 5) Please note that all corresponding authors are required to supply an ORCID ID for their name upon submission of a revised manuscript.
- 6) It is mandatory to include a 'Data Availability' section after the Materials and Methods. Before submitting your revision, primary datasets produced in this study need to be deposited in an appropriate public database, and the accession numbers and

database listed under 'Data Availability'. Please remember to provide a reviewer password if the datasets are not yet public (see <https://www.embopress.org/page/journal/17574684/authorguide#dataavailability>).

12) Author contributions: You will be asked to provide CRediT (Contributor Role Taxonomy) terms in the submission system. These replace a narrative author contribution section in the manuscript.

13) A Conflict of Interest statement should be provided in the main text.

14) Every published paper now includes a 'Synopsis' to further enhance discoverability. Synopses are displayed on the journal webpage and are freely accessible to all readers. They include a short stand first (maximum of 300 characters, including space) as well as 2-5 one-sentences bullet points that summarizes the paper. Please write the bullet points to summarize the key NEW findings. They should be designed to be complementary to the abstract - i.e. not repeat the same text. We encourage inclusion of key acronyms and quantitative information (maximum of 30 words / bullet point). Please use the passive voice. Please attach these in a separate file or send them by email, we will incorporate them accordingly.

15) Include a Reagents and Tools Table as part of the Methods section, which can be downloaded from our author guidelines (<https://www.embopress.org/page/journal/17574684/authorguide#structuredmethods>)

***** Reviewer's comments *****

Referee #1 (Comments on Novelty/Model System for Author):

In this study, Zhao et al. characterized monoclonal antibodies (MAbs) targeting M1R and B6R of monkeypox virus (MPXV) for therapeutic development. Several broadly effective neutralizing MAbs were identified, which exhibited synergistic antiviral effects in antibody cocktail and bispecific antibody designs. Notably, bispecific antibodies showed robust protective efficacy against vaccinia virus in a mouse model, highlighting their potential as therapeutic candidates against MPXV and other orthopoxvirus infections. Overall, the research is detailed and thorough.

Referee #1 (Remarks for Author):

In this study, Zhao et al. characterized monoclonal antibodies (MAbs) targeting M1R and B6R of monkeypox virus (MPXV) for therapeutic development. Several broadly effective neutralizing MAbs were identified, which exhibited synergistic antiviral effects in antibody cocktail and bispecific antibody designs. Notably, bispecific antibodies showed robust protective efficacy against vaccinia virus in a mouse model, highlighting their potential as therapeutic candidates against MPXV and other orthopoxvirus infections. Overall, the research is detailed and thorough. This reviewer has raised the following concerns, hoping they will be helpful in improving the article.

Major Points:

1. In Figure 2, since the MAbs in this study were derived from MPXV antigen-immunized mice, why did the authors only measure PRNT50 for VACV instead of MPXV neutralization? Could an explanation be provided?
2. In Figure 3, the dosing timepoints were set at 4 hours pre- and post-challenge. Given that antibody half-lives typically last up to two weeks, why were two doses administered within such a short window? Was this testing prophylactic or therapeutic protection?
3. For the viral load assays in Fig. 3J and 3M, on which day post-challenge were samples from the Ctr mAb and PBS groups collected? Survival data indicate these negative controls died by day 5.
4. In Figure 4, mMM1-16 showed suboptimal neutralizing activity. Why was structural biology performed on this antibody instead of mMM1-39 or mMM1-40, which have better neutralizing activity? While their epitopes compete with 7D11, they may not fully overlap.
5. Figures 5D-5F lack significant insights and could be moved to supplementary materials. Additionally, why were B6 antibodies discovered in this study (e.g., mMB6-18 or mMB6-23) not combined with mMM1-40 in a cocktail? Could the authors clarify?

Minor points :

1. Since M1R and B6R are distinct targets (not different epitopes on the same target), the rationale for bispecific antibody design is insufficiently justified. Could the authors elaborate?
2. Prior studies report that 10% complement addition inhibits viral plaques. The authors should confirm whether complement itself interferes with VACV infection in their experimental system.
3. The antibody sequences obtained in this study should be provided in supplementary information to facilitate reproducibility and follow-up research.

Referee #2 (Comments on Novelty/Model System for Author):

1. Technical Quality: Medium. I think the description of the results should be more detailed and in line with the data; At the same time, the discussion section should carefully explore the role of this article in promoting existing research. Humanization and bispecific modification of mMM1-40 were carried out in the article, and the protective effect was verified; and the structural basis

of mMM1-40 should be analyzed. On the other hand, in the experiment of the protective effect of bispecific antibodies, the animal monitoring time after administration was not long enough.

2. Novelty: High. The study innovatively engineered a bispecific antibody by integrating M1R and B6R antibodies into a single molecular entity, enabling simultaneous targeting of two distinct viral particles of monkeypox virus. The protective efficacy of these antibodies was rigorously assessed through comprehensive evaluation using multiple established MPXV (CB-17 SCID and CAST/EiJ) and VACV (BALB/c) murine infection models.

3. Medical impact: High. This study aligns with the current research context of the monkeypox pandemic, demonstrating that the developed antibodies and bispecific antibodies exhibit significant protective efficacy in both VACV and MPXV infection models, indicating promising potential for clinical applications.

4. Adequacy of model system: Adequate. This study conducted systematic screening of murine-derived antibodies, comprehensively assessing their binding affinity and neutralizing capacity. The protective efficacy was further validated through well-established VACV and MPXV murine challenge models, demonstrating methodological rigor and scientific completeness throughout the investigation.

Referee #2 (Remarks for Author):

In the manuscript entitled "Anti-M1R/B6R antibody characterization and bispecific design for enhanced orthopoxvirus protection", Zhao et al. systematically characterize the binding affinity and neutralizing capacity of antibodies targeting monkeypox virus proteins M1R and B6R, while evaluating their protective efficacy against both MPXV and VACV infections in murine models. The data is solid and the MS is well written. Considering the MPXV outbreak is still ongoing worldwide, the work is of great interest to the field. However, I have some concerns to be addressed before publication.

1. In Figure 1, the authors classified the epitopes of Mabs using 32 peptides, and what are the methodologies employed for the construction, expression, and functional/structural validation of the 32 peptides?
2. The study initially mapped the binding epitopes of monkeypox virus M1R and B6R antibodies, subsequently classifying these antibodies into distinct groups according to their epitope recognition patterns. This raises the question: what are the differential characteristics in terms of binding affinity and neutralizing capacity among these epitope-defined antibody groups?
3. In the VACV infection in vivo experiment, antibodies targeting M1R, including mMM1-16, mMM1-40 and hmMM1-40 were selected. Compared to mMM1-39 and mMM1-40, mMM1-16 demonstrates inferior neutralizing activity, with an PRNT50 exceeding 100 $\mu\text{g}/\text{mL}$. Notably, while treatment with mMM1-16 did not significantly reduce lung viral load compared to PBS controls, it effectively prevented substantial weight loss in infected mice. What potential mechanisms could account for this observed dissociation between virological and clinical outcomes?
4. Given that the authors monitored both body weight changes and viral load in mice for extended periods (5-9 days) post-viral challenge, did they further prolong the antibody administration period to evaluate its protective effects?
5. What is the structural basis underlying mMM1-40's high-affinity binding to M1R, given its demonstrated efficacy in both in vitro neutralization assays and in vivo protection studies, as well as its successful incorporation into a bispecific antibody with hMB621?
6. Is pulmonary viral load in mice post-viral challenge a determinant of lethal outcome?
7. Overall, the time period for all challenge experiments is not long enough. In Figures 3 and 5, weight and survival monitoring only lasted for a maximum of ten days while only 5 days were monitored in Figure 6, and I think it is not long enough, especially when each mouse was only given 5mg/kg. Will extending the monitoring time lead to a decrease in survival rate?
8. A comprehensive discussion section is needed, particularly focusing on: (1) the current research landscape regarding M1R and B6R antibodies, and (2) the study's substantive contributions to advancing this field of research.

We require:

Response: Provided.

2) Individual production quality figure files as .eps, .tif, .jpg (one file per figure). For guidance, download the 'Figure Guide PDF': (<https://www.embopress.org/page/journal/17574684/authorguide#figureformat>).

Response: Done.

3) A .docx formatted letter INCLUDING the reviewers' reports and your detailed point-by-point responses to their comments. As part of the EMBO Press transparent editorial process, the point-by-point response is part of the Review Process File (RPF), which will be published alongside your paper.

Response: Provided.

4) A complete author checklist, which you can download from our author guidelines (<https://www.embopress.org/page/journal/17574684/authorguide#submissionofrevisions>). Please insert information in the checklist that is also reflected in the manuscript. The completed author checklist will also be part of the RPF.

Response: Provided.

Response: Provided.

6) It is mandatory to include a 'Data Availability' section after the Materials and Methods. Before submitting your revision, primary datasets produced in this study need to be deposited in an appropriate public database, and the accession numbers and database listed under 'Data Availability'. Please remember to provide a reviewer password if the datasets are not yet public (see <https://www.embopress.org/page/journal/17574684/authorguide#dataavailability>).

Response: Included.

7) For data quantification: please specify the name of the statistical test used to generate error bars and P values, the number (n) of independent experiments (specify technical or biological

replicates) underlying each data point and the test used to calculate p-values in each figure legend. The figure legends should contain a basic description of n, P and the test applied. Graphs must include a description of the bars and the error bars (s.d., s.e.m.). See also 'Figure Legend' guidelines: <https://www.embopress.org/page/journal/17574684/authorguide#figureformat>

Response: Done.

Response: Done.

9) Our journal encourages inclusion of *data citations in the reference list* to directly cite datasets that were re-used and obtained from public databases. Data citations in the article text are distinct from normal bibliographical citations and should directly link to the database records from which the data can be accessed. In the main text, data citations are formatted as follows: "Data ref: Smith et al, 2001" or "Data ref: NCBI Sequence Read Archive PRJNA342805, 2017". In the Reference list, data citations must be labeled with "[DATASET]". A data reference must provide the database name, accession number/identifiers and a resolvable link to the landing page from which the data can be accessed at the end of the reference. Further instructions are available at <https://www.embopress.org/page/journal/17574684/authorguide#referencesformat>.

Response: Not involve.

10) We replaced Supplementary Information with Expanded View (EV) Figures and Tables that are collapsible/expandable online. A maximum of 5 EV Figures can be typeset. EV Figures should be cited as 'Figure EV1, Figure EV2" etc... in the text and their respective legends should be included in the main text after the legends of regular figures.- For the figures that you do NOT wish to display as Expanded View figures, they should be bundled together with their legends in a single PDF file called *Appendix*, which should start with a short Table of Content. Appendix figures should be referred to in the main text as: "Appendix Figure S1, Appendix Figure S2" etc.

<https://www.embopress.org/page/journal/17574684/authorguide#expandedview>.

Response: Four EV figures, 13 appendix figures and 5 appendix tables are included in our manuscript.

- the medical issue you are addressing,
- the results obtained and

- their clinical impact.

Response: The relevant information is provided as follows.

The paper explained

Problem

Mpox, caused by the mpox virus (MPXV), poses a significant threat to global public health. However, effective and specific therapeutic strategies remain lacking. Urgent efforts are needed to develop antibody-based therapeutics.

Results

We dissected the epitope characteristics of MPXV MIR and B6R by characterizing a panel of monoclonal antibodies (MAbs). Several broadly neutralizing anti-MIR and anti-B6R MAbs were identified and they exhibited enhanced antiviral effects against MPXV when used in antibody cocktail. Additionally, we explored bispecific antibody designs and found the VH-CHI switch region-inserting format exhibited robust protective efficacy against VACV in a mouse model.

Impact

Our findings underscore the therapeutic potential of MIR- and B6R- based approaches and provide promising antibody candidates against MPXV and other orthopoxviruses.

12) Author contributions: You will be asked to provide CRediT (Contributor Role Taxonomy) terms in the submission system. These replace a narrative author contribution section in the manuscript.

Response: Provided.

13) A Conflict of Interest statement should be provided in the main text.

Response: Provided.

14) Every published paper now includes a 'Synopsis' to further enhance discoverability. Synopses are displayed on the journal webpage and are freely accessible to all readers. They include a short stand first (maximum of 300 characters, including space) as well as 2-5 one-sentences bullet points that summarizes the paper. Please write the bullet points to summarize the key NEW findings. They should be designed to be complementary to the abstract - i.e. not repeat the same text. We encourage inclusion of key acronyms and quantitative information (maximum of 30 words / bullet point). Please use the passive voice. Please attach these in a separate file or send them by email, we will incorporate them accordingly.

Response: Provided.

guidelines

***** Reviewer's comments *****

Referee #1 (Remarks for Author):

In this study, Zhao et al. characterized monoclonal antibodies (MAbs) targeting M1R and B6R of monkeypox virus (MPXV) for therapeutic development. Several broadly effective neutralizing MAbs were identified, which exhibited synergistic antiviral effects in antibody cocktail and bispecific antibody designs. Notably, bispecific antibodies showed robust protective efficacy against vaccinia virus in a mouse model, highlighting their potential as therapeutic candidates against MPXV and other orthopoxvirus infections. Overall, the research is detailed and thorough. This reviewer has raised the following concerns, hoping they will be helpful in improving the article.

Response: We thank you for your positive comments and constructive suggestions to our manuscript.

Major Points:

1. In Figure 2, since the MAbs in this study were derived from MPXV antigen-immunized mice, why did the authors only measure PRNT50 for VACV instead of MPXV neutralization? Could an explanation be provided?

Response: The MAbs in our study were indeed generated from mice immunized with MPXV antigens. Given the high antigenic similarity between MPXV and VACV, our binding assays demonstrated that these MAbs exhibited cross-reactivity with antigenic orthologs from both viruses. Additionally, MPXV assays require a biosafety level 3 (BSL-3) laboratory environment, whereas VACV assays can be conducted under BSL-2 conditions. Considering the limited availability of BSL-3 facilities, we employed VACV as a surrogate in the neutralizing assay.

2. In Figure 3, the dosing timepoints were set at 4 hours pre- and post-challenge. Given that antibody half-lives typically last up to two weeks, why were two doses administered within such a short window? Was this testing prophylactic or therapeutic protection?

Response: This experiment aimed to preliminarily screen MAbs for their protective potential in a murine model, evaluating both prophylactic and early therapeutic effects. Specifically, the first dose, administered 4 hours before viral challenge, aimed to assess prophylactic protection, while the second dose, given 4 hours after challenge, was intended to simulate early post-exposure treatment. Although IgG antibodies generally have long half-lives in vivo, we selected this short interval, dual-dosing strategy to ensure sufficient antibody levels during the critical window of early infection.

3. For the viral load assays in Fig. 3J and 3M, on which day post-challenge were samples from the Ctr mAb and PBS groups collected? Survival data indicate these negative controls died by day 5.

Response: We apologize for the omission in the figure legends. In Figures 3J and 3M, samples from the PBS and Ctr MAb groups were collected on day 5 post-challenge, as these mice met the humane endpoint criteria (>20% weight loss) and were euthanized. These control groups served as positive controls for viral load quantification in lungs. The figure legends have been updated to include this information.

4. In Figure 4, mMM1-16 showed suboptimal neutralizing activity. Why was structural biology performed on this antibody instead of mMM1-39 or mMM1-40, which have better neutralizing activity? While their epitopes compete with 7D11, they may not fully overlap.

Response: We agree that selecting the functionally potent antibodies is of utmost importance for structural characterization. After receiving your suggestion, we immediately prepared the hmMM1-40 Fab and MPXV MIR complex proteins and performed crystallization screening assays. We successfully resolved the crystal structure of the complex at a resolution of 2.8 Å. We compared the binding characteristics of hmMM1-40 and 7D11 and reprepared Figure 4, as shown below. A detailed description is provided in the manuscript in lines 284-308.

Furthermore, we performed alanine-scanning mutagenesis on the MIR epitope residues involved in hydrogen bond interaction to assess their impact on antibody-antigen binding. The results have been incorporated into the manuscript in lines 310-318, and the corresponding data are presented in Figure EV2, as shown below.

Figure 4

Figure EV2

5. Figures 5D-5F lack significant insights and could be moved to supplementary materials. Additionally, why were B6 antibodies discovered in this study (e.g., mMB6-18 or mMB6-23) not combined with mMM1-40 in a cocktail? Could the authors clarify?

Response: Figure 5D–5F have been moved to the supplementary materials as Figure EV3, with the updated Figure 5 is shown below. Regarding the antibody cocktail design, hMB621 (a fully human antibody reported in our previous paper (DOI:10.1038/s41467-024-48312-2) was selected as the combination partner with mMM1-40 because it displayed the superior in vitro neutralizing activity compared to the B6R antibodies isolated in this study. Additionally, since our goal is therapeutic development for human use, hMB621's fully human origin aligns with this objective, similar to our humanization of mMM1-40. While the B6R antibodies in this study showed promising binding, their neutralization efficacy did not surpass hMB621, making the latter a more optimal candidate for combinatorial therapy.

Minor points :

1. Since M1R and B6R are distinct targets (not different epitopes on the same target), the rationale for bispecific antibody design is insufficiently justified. Could the authors elaborate?

Response: Bispecific antibodies can be designed to target different epitopes on the same antigen or epitopes on different antigens. Our rationale for designing bispecific antibodies targeting M1R and B6R was based on their complementary roles in neutralizing both forms of orthopoxvirus particles—M1R being a major component of the intracellular mature virion (IMV), and B6R serving as a key component of the extracellular enveloped virion (EEV). Previous studies have demonstrated that simultaneous targeting of IMV and EEV antigens can enhance antiviral efficacy and reduce the risk of viral escape. Therefore, we aimed to engineer a bispecific antibody capable of engaging both targets to achieve broader and more effective neutralization.

2. Prior studies report that 10% complement addition inhibits viral plaques. The authors should confirm whether complement itself interferes with VACV infection in their experimental system.

Response: In our neutralization assays, complement was only used in the neutralization of anti-B6R antibodies against EEV. To address your concern, we conducted additional experiments to evaluate whether complement alone affects EEV infection. We tested three concentrations of complement for their effects on EEV-induced plaque formation. The results showed that complement alone did not significantly impact plaque numbers. The data is shown below and have

been added as Appendix Figure S11. The corresponding descriptions have been added to the manuscript in lines 218-220. Importantly, no complement was used in the neutralization assays of M1R-targeting antibodies against IMV, ensuring that the observed neutralizing effects were not influenced by complement.

3. The antibody sequences obtained in this study should be provided in supplementary information to facilitate reproducibility and follow-up research.

Response: *The germline assignments and CDR3 amino acid sequences of both heavy and light chains have been provided in the Appendix Table S1.*

Referee #2 (Remarks for Author):

In the manuscript entitled "Anti-M1R/B6R antibody characterization and bispecific design for enhanced orthopoxvirus protection", Zhao et al. systematically characterize the binding affinity and neutralizing capacity of antibodies targeting monkeypox virus proteins M1R and B6R, while evaluating their protective efficacy against both MPXV and VACV infections in murine models. The data is solid and the MS is well written. Considering the MPXV outbreak is still ongoing worldwide, the work is of great interest to the field. However, I have some concerns to be addressed before publication.

Response: *We sincerely appreciate your positive comments and thoughtful suggestions. These insights are very helpful and have significantly contributed to the improvement of our work.*

1. In Figure 1, the authors classified the epitopes of Mabs using 32 peptides, and what are the methodologies employed for the construction, expression, and functional/structural validation of the 32 peptides?

Response: *The peptides used for epitope mapping in Figure 1 were chemically synthesized by a commercial provider. Each peptide was purified by high-performance liquid chromatography (HPLC) and validated by mass spectrometry (MS) analysis to confirm sequence integrity and molecular weight.*

2. The study initially mapped the binding epitopes of monkeypox virus M1R and B6R antibodies,

subsequently classifying these antibodies into distinct groups according to their epitope recognition patterns. This raises the question: what are the differential characteristics in terms of binding affinity and neutralizing capacity among these epitope-defined antibody groups?

Response: *Following your good suggestion, we performed correlation analyses between the binding affinity (K_D) to the VACV B5R and the neutralizing activity ($PRNT_{50}$) against VACV, according to epitope groups. Among the anti-B6R antibodies, only those recognizing the SCR1–2 region exhibited a positive correlation between binding affinity and neutralizing activity, and no such correlation was observed in other groups. The results are presented below and included in the manuscript as Figure EV1.*

For anti-M1R antibodies, a visible positive correlation between binding affinity and neutralization potency was observed in two neutralizing antibody groups: those recognizing a conformational neutralizing epitope (mMM1-39 and mMM1-40) and those targeting a linear neutralizing epitope (mMM1-10 and mMM1-16). No correlation was observed among non-neutralizing antibodies. The corresponding descriptions have been added to the manuscript in lines 202-205 and 216-218.

3. In the VACV infection in vivo experiment, antibodies targeting M1R, including mMM1-16, mMM1-40 and hmMM1-40 were selected. Compared to mMM1-39 and mMM1-40, mMM1-16 demonstrates inferior neutralizing activity, with an $PRNT_{50}$ exceeding 100 $\mu\text{g}/\text{mL}$. Notably, while treatment with mMM1-16 did not significantly reduce lung viral load compared to PBS controls, it effectively prevented substantial weight loss in infected mice. What potential mechanisms could account for this observed dissociation between virological and clinical outcomes?

Response: *In our neutralization assays, we found that mMM1-16 exhibited a certain degree of neutralization (Appendix Figure S10), but was only able to achieve 50% neutralization at tested*

antibody concentration. Therefore, its neutralizing activity was shown as exceeding 100 µg/mL. In our 5-day monitoring animal experiment (Figure 3B-C), mice treated with mMM1-16 exhibited partially alleviated weight loss compared to the PBS and control antibody groups. However, in the extended 9-day monitoring experiment (Figure 3H-I), body weight in the mMM1-16 group dropped below 80% of the initial weight by Day 6, with no substantial protection was observed thereafter. These results indicate that the protective effect of mMM1-16 is limited, which is consistent with its weak *in vitro* neutralizing activity.

4. Given that the authors monitored both body weight changes and viral load in mice for extended periods (5-9 days) post-viral challenge, did they further prolong the antibody administration period to evaluate its protective effects?

Response: Following your valuable suggestion, we immediately performed *in vivo* experiments to assess the protective efficacy of antibody over an extended administration window, in which mice were administered antibodies either 24 hours before (-24h) or 24 hours after (+24h) viral challenge, representing prophylactic and therapeutic settings, respectively. Three antibody groups were evaluated: the cocktail (hmMM1-40+hMB621) and two bispecific antibodies (hmMM1-40-SW-hMB621 and hMB621-SW-hmMM1-40). Each group was administered intraperitoneally at a dose of 5 mg/kg, and mice were monitored for 9 days post-infection to assess body weight and lung viral load. We found that both the antibody cocktail and bispecific antibodies conferred effective protection in both -24h and +24h treatment conditions. The data are shown below and have been included in the manuscript as Figure EV4. The corresponding descriptions have been incorporated into the manuscript in lines 393-403.

5. What is the structural basis underlying mMM1-40's high-affinity binding to M1R, given its demonstrated efficacy in both *in vitro* neutralization assays and *in vivo* protection studies, as well as its successful incorporation into a bispecific antibody with hMB621?

Response: We agree that revealing the structural basis of mMM1-40 is needed due to its effective antiviral activity. After receiving your suggestion, we immediately prepared the hmMM1-40 Fab

and MPXV MIR complex proteins and performed crystallization screening assays. Fortunately, we successfully resolved the crystal structure of the complex at a resolution of 2.8 Å. We compared the binding characteristics of hmMM1-40 and 7D11 and repaired Figure 4, as shown below. A detailed description is provided in the manuscript in lines 284-308.

Furthermore, we performed alanine-scanning mutagenesis on the MIR epitope residues involved in hydrogen bond interaction to assess their impact on antibody-antigen binding. The results have been incorporated into the manuscript in lines 310-318, and the corresponding data are presented in Figure EV2, as shown below.

Figure 4

Figure EV2

6. Is pulmonary viral load in mice post-viral challenge a determinant of lethal outcome?

Response: According to previous studies, VACV or MPXV infection model generally involved lower respiratory tract infection and systemic dissemination infection, as reported by Gilchuk et al. in 2016 Cell publication (DOI: 10.1016/j.cell.2016.09.049). In the lower respiratory tract infection model, mice were challenged intranasally, and viral loads were tested in the lungs. In our study, the viruses were challenged either intranasally or intraperitoneally, and we measured the viral loads in lung (Figure 3 and 6) or in the blood and other organs, including lung, spleen, ovary/testicle, liver, kidney and brain (Figure 5).

7. Overall, the time period for all challenge experiments is not long enough. In Figures 3 and 5, weight and survival monitoring only lasted for a maximum of ten days while only 5 days were monitored in Figure 6, and I think it is not long enough, especially when each mouse was only given 5mg/kg. Will extending the monitoring time lead to a decrease in survival rate?

Response: Thank you for your comment. In our in vivo challenge experiments, we observed that the mice typically succumbed on Day 5 or 6 after intranasal or intraperitoneal challenge with a

lethal dose (Figure 3, 5 and 6). By Day 9 or 10, surviving mice generally regained their body weight to 100% of the initial level. The shortest monitoring windows were therefore designed according to the time of death of the control groups. As shown in Major Point 4, we extended the monitoring time for the antibody cocktail and bispecific antibodies previously presented in Figure 6. We similarly observed that surviving mice regained their body weight to 100% of the initial level by Day 9.

8. A comprehensive discussion section is needed, particularly focusing on: (1) the current research landscape regarding M1R and B6R antibodies, and (2) the study's substantive contributions to advancing this field of research.

Response: *Thank you for your good suggestion. We have added the relevant discussion in the Discussion section.*

23rd Jul 2025

Dear Prof. Gao,

Thank you for the submission of your revised manuscript to EMBO Molecular Medicine. I am pleased to inform you that we will be able to accept your manuscript pending the following final amendments:

1) In the main manuscript file, please do the following:

- Please address all comments suggested by our data editors listed below:

o Data availability statement:

1. Please note that the specific URLs for 9VHZ, 9LF8 datasets are not provided in the data availability statement.

o Figure legends:

1. Please note that the exact p values are not provided in the legends of figures 3D, G, J, M; 5C, E; 6D, EV2 B, EV3 C.

2. Please note that information related to n is missing in the legends of figures 3B, D, E, G, H, J, K, M; 5C, 6D, EV2 B, EV3 C, EV4 C, F.

3. Please note that the error bars are not defined in the legends of figures 3B, D, E, G, H, J, K, M; 5A, C, D, E; 6B, D; EV2 B, EV3 A, C; EV4 A, C, D, F.

- Limit Keywords to max. 5.

- Figure callouts should be in a sequential order. Currently, Fig 2A is called out before Fig 1C. Please correct. Also, please add callouts for for Fig 4C, Appendix Table S1. Fig EV2 A and B and Fig EV3 A, B, and C.

- Rename "Material and Methods" to "Methods".

- Rename "Conflict of interest" to "Disclosure and competing interests statement". We updated our journal's competing interests policy in January 2022 and request authors to consider both actual and perceived competing interests. Please review the policy <https://www.embopress.org/competing-interests> and update your competing interests if necessary.

- Indicate in legends exact n and exact p values, not a range, along with the statistical test used. To keep the figures "clear" some authors found providing an Appendix table Sx with all exact p-values preferable. You are welcome to do this if you want to.

- In Methods, a statistical paragraph should reflect all information that you have filled in the Authors Checklist, especially regarding randomization, blinding, replication.

- Please use the following format to report the accession number of your data:

[data type]: [full name of the resource] [accession number/identifier] ([doi or URL or identifiers.org/DATABASE:ACCESSION])

Please check "Author Guidelines" for more information.

<https://www.embopress.org/page/journal/17574684/authorguide#availabilityofpublishedmaterial>

2) Appendix: Please remove line numbers and add a table of contents with page numbers on the title page.

3) Source data: Please upload the source data files as one (ZIP) file per figure and upload a completed source data checklist.

4) The Paper Explained: Please provide "The Paper Explained" and add it to the main manuscript text. Please check "Author Guidelines" for more information. <https://www.embopress.org/page/journal/17574684/authorguide#researcharticleguide>

5) Synopsis:

- Synopsis image: Please resize the image to 550 px-wide x 300-600 pixels high and upload it as a high-resolution jpeg file.

6) As part of the EMBO Publications transparent editorial process initiative (see our Editorial at

<http://embomolmed.embopress.org/content/2/9/329>), EMBO Molecular Medicine will publish online a Review Process File (RPF) to accompany accepted manuscripts. This file will be published in conjunction with your paper and will include the anonymous referee reports, your point-by-point response and all pertinent correspondence relating to the manuscript. Let us know whether you agree with the publication of the RPF and as here, if you want to remove or not any figures from it prior to publication.

7) Please provide a point-by-point letter INCLUDING my comments as well as the reviewer's reports and your detailed responses (as Word file).

I look forward to reading a new revised version of your manuscript as soon as possible.

Yours sincerely,

Zeljko Durdevic

Zeljko Durdevic
Senior Editor

*** Instructions to submit your revised manuscript ***

- 1) a .docx formatted version of the manuscript text (including Figure legends and tables)
 - 2) Separate figure files*
 - 3) supplemental information as Expanded View and/or Appendix. Please carefully check the authors guidelines for formatting Expanded view and Appendix figures and tables at <https://www.embopress.org/page/journal/17574684/authorguide#expandedview>
 - 4) a letter INCLUDING the reviewer's reports and your detailed responses to their comments (as Word file).
 - 5) The paper explained: EMBO Molecular Medicine articles are accompanied by a summary of the articles to emphasize the major findings in the paper and their medical implications for the non-specialist reader. Please provide a draft summary of your article highlighting
 - the medical issue you are addressing,
 - the results obtained and
 - their clinical impact.This may be edited to ensure that readers understand the significance and context of the research. Please refer to any of our published articles for an example.
 - 6) Author contributions: the contribution of every author must be detailed in a separate section.
 - 7) EMBO Molecular Medicine now requires a complete author checklist (<https://www.embopress.org/page/journal/17574684/authorguide>) to be submitted with all revised manuscripts. Please use the checklist as guideline for the sort of information we need WITHIN the manuscript. The checklist should only be filled with page numbers where the information can be found. This is particularly important for animal reporting, antibody dilutions (missing) and exact values and n that should be indicated instead of a range.
 - 8) Every published paper now includes a 'Synopsis' to further enhance discoverability. Synopses are displayed on the journal webpage and are freely accessible to all readers. They include a short stand first (maximum of 300 characters, including space) as well as 2-5 one sentence bullet points that summarise the paper. Please write the bullet points to summarise the key NEW findings. They should be designed to be complementary to the abstract - i.e. not repeat the same text. We encourage inclusion of key acronyms and quantitative information (maximum of 30 words / bullet point). Please use the passive voice. Please attach these in a separate file or send them by email, we will incorporate them accordingly.
- You are also welcome to suggest a striking image or visual abstract to illustrate your article. If you do please provide a jpeg file 550 px-wide x 300-600px high.
- 9) A Conflict of Interest statement should be provided in the main text
 - 10) Please note that we now mandate that all corresponding authors list an ORCID digital identifier. This takes <90 seconds to complete. We encourage all authors to supply an ORCID identifier, which will be linked to their name for unambiguous name

identification.

Currently, our records indicate that the ORCID for your account is 0000-0002-3869-615X.

Link Not Available

11) Include a Reagents and Tools Table as part of the Methods section, which can be downloaded from our author guidelines (<https://www.embopress.org/page/journal/17574684/authorguide#structuredmethods>)

Photos 400-800 DPI

*Additional important information regarding figures and illustrations can be found at

<https://bit.ly/EMBOPressFigurePreparationGuideline>. See also figure legend preparation guidelines:

<https://www.embopress.org/page/journal/17574684/authorguide#figureformat>

***** Reviewer's comments *****

Referee #1 (Comments on Novelty/Model System for Author):

The mouse model infected with vaccinia virus is suitable for evaluating the protective effect of neutralizing antibodies.

Referee #1 (Remarks for Author):

The questions raised by this reviewer have been well supplemented and explained, and there are no more questions.

Referee #2 (Comments on Novelty/Model System for Author):

1 . Technical Quality: High. This manuscript provides a comprehensive account of the study's significant contributions to monkeypox antibody research. It systematically addresses reviewer comments, presents the resolved structure of the mMM1-40-M1R complex, and includes enhanced data from animal experiments.

2 . Novelty: High. The research innovatively designed a bispecific antibody by integrating the M1R and B6R antibodies into a single molecular entity, enabling simultaneous targeting of two distinct virions of the monkeypox virus. Researchers conducted a thorough evaluation of the protective efficacy of these antibodies using well-established infection models for MPXV (CB-17 SCID and CAST/EiJ mice) and VACV (BALB/c mice).

3 . Medical impact: High. Grounded in the context of the ongoing monkeypox outbreak, the results demonstrate that both the developed monoclonal antibodies and the bispecific antibody confer significant protection in both VACV and MPXV infection models, highlighting their considerable clinical potential.

4 . Adequacy of model system: Adequate. The study involved systematic screening of murine-derived antibodies, comprehensive assessment of their binding affinity and neutralizing capacity, and rigorous validation of protective efficacy through established VACV and MPXV murine challenge models. The overall research methodology is scientifically rigorous and comprehensive.

Referee #2 (Remarks for Author):

The study encompassed a systematic screening pipeline for murine-derived antibodies, leveraging established techniques to identify promising candidates. This was followed by a comprehensive in vitro characterization, including meticulous assessment of binding affinity, and neutralizing capacity against relevant viral strains. Subsequently, the protective efficacy of the lead antibodies was subjected to rigorous in vivo validation employing well-established murine challenge models for both Vaccinia

virus (VACV) in BALB/c mice and Monkeypox virus (MPXV) in CB-17 SCID and CAST/EiJ mouse strains. This multi-faceted approach, integrating systematic discovery, detailed functional profiling, and robust in vivo efficacy testing across relevant models, underscores the scientific rigor and comprehensive nature of the overall research methodology. I consider this manuscript well-positioned for publication in EMBO Molecular Medicine in its current form, based on its scientific rigor and potential impact.

The authors addressed the remaining editorial issues.

15th Aug 2025

Dear Prof. Gao,

We are pleased to inform you that your manuscript is accepted for publication and is now being sent to our publisher to be included in the next available issue of EMBO Molecular Medicine.

Zeljko Durdevic
Senior Editor
EMBO Molecular Medicine
